# The World Is Bigger!
# A Computationally-Embedded Perspective
# on the Big World Hypothesis

**Alex Lewandowski**[1,2,*]    **Aditya A. Ramesh**[3]    **Edan Meyer**[1,2]
**Dale Schuurmans**[1,2,4,5]    **Marlos C. Machado**[1,2,4]
[1]University of Alberta    [2]Amii    [3]The Swiss AI Lab IDSIA, USI & SUPSI
[4]Canada CIFAR AI Chair    [5]Google DeepMind

## Abstract

Continual learning is often motivated by the idea, known as the big world hypothesis, that "the world is bigger" than the agent. Recent problem formulations capture this idea by explicitly constraining an agent relative to the environment. These constraints lead to solutions in which the agent continually adapts to best use its limited capacity, rather than converging to a fixed solution. However, explicit constraints can be ad hoc, difficult to incorporate, and may limit the effectiveness of scaling up the agent's capacity. In this paper, we characterize a problem setting in which an agent, regardless of its capacity, is constrained by being embedded in the environment. In particular, we introduce a *computationally-embedded* perspective that represents an embedded agent as an automaton simulated within a universal (formal) computer. Such an automaton is always constrained; we prove that it is equivalent to an agent that interacts with a partially observable Markov decision process over a countably infinite state-space. We propose an objective for this setting, which we call *interactivity*, that measures an agent's ability to continually adapt its behaviour by learning new predictions. We then develop a model-based reinforcement learning algorithm for interactivity-seeking, and use it to construct a synthetic problem to evaluate continual learning capability. Our results show that deep nonlinear networks struggle to sustain interactivity, whereas deep linear networks sustain higher interactivity as capacity increases.

## 1   Introduction

The goal of this paper is to characterize a general problem setting in which the best use of an agent's limited capacity is to continually adapt (Ring, 1994; Thrun, 1998; Abel et al., 2023). Our approach is motivated by the idea, known as the big world hypothesis, that "the world is bigger" than the agent (Javed and Sutton, 2024). That is, an agent in a big world may lack the capacity to learn the fixed optimal solution, and should instead continually adapt to new experience by updating its approximate solution (*i.e.*, by tracking, Sutton et al., 2007). However, formalizing the relationship between the agent and the environment presents a challenge, because they are typically treated as separate entities in reinforcement learning (see Figures 1a and 1b). We address this by introducing a computationally-embedded agent that is simulated by the environment's dynamics, thus constraining the agent within the environment. With this perspective, we construct a problem setting in which any such agent is (i) constrained by its capacity, and (ii) suboptimal if it stops learning.

Explicit constraints on the agent have been previously considered in continual learning as a means of capturing the big world hypothesis. For example, in continual learning experiments, the agent's

---

*Correspondence to: Alex Lewandowski <lewandowski@ualberta.ca>.

39th Conference on Neural Information Processing Systems (NeurIPS 2025).

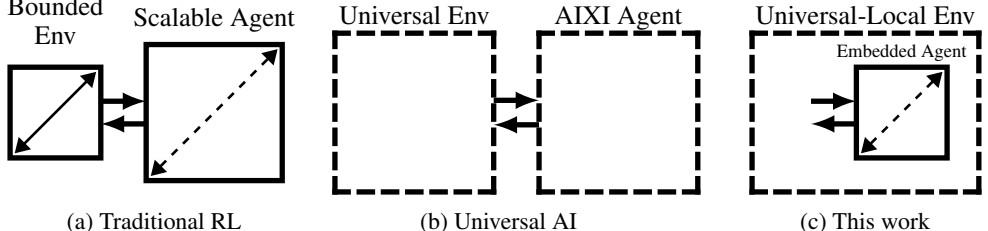

(a) Traditional RL                    (b) Universal AI                    (c) This work

Figure 1: **Comparing the agent and environment in different problem formulations.** Each problem formulation differs in the constraints that it imposes on the agent relative to the environment. **Traditional RL:** A given environment typically has a bounded capacity, but a scalable agent can increase its capacity. Such an agent is unconstrained in principle because its capacity can always be scaled beyond the environment. **Universal AI:** Both the computationally universal environment and the AIXI agent are unbounded. AIXI is thus unconstrained but not computable. **This work:** The universal-local environment is unbounded, but it can simulate an embedded agent of any bounded capacity within its state-space. An embedded agent is implicitly constrained because the environment necessarily has greater capacity than any agent contained within it.

learning algorithm is often explicitly constrained by what it can store (Prabhu et al., 2020), or by the expressivity of its function approximator (Meyer et al., 2024). Other real-world constraints on the agent's hardware have also been considered. Such constraints include limits on the agent's compute (see discussion on measuring compute in Section 4.1, Verwimp et al., 2024) and on the agent's energy use (Javed and Sutton, 2024). One recent formalization of continual learning uses an explicit constraint on the agent's information-theoretic capacity (Kumar et al., 2023, 2024). However, beyond analytically tractable agents and environments, this information-theoretic constraint is difficult to measure and enforce, limiting its generality as a problem setting for continual learning. Furthermore, explicitly constraining the agent limits the effectiveness of scaling up the agent's capacity, which has been a major source of progress in machine learning more broadly (Hestness et al., 2017; Kaplan et al., 2020; Hoffmann et al., 2022). These limitations suggest that explicit constraints may not be an effective way of capturing the big world hypothesis.

In contrast to explicit constraints, our approach considers the implicit constraint that arises from an agent embedded in an environment (see Figure 1c). Specifically, an embedded agent is, in principle, fully defined within the environment's state, and simulated by the environment's dynamics. The embedded nature of agents—that they exist within and as part of their environment—is typically treated as outside the scope of the problem formulation (Demski and Garrabrant, 2019). However, the physical world is a clear example of a world bigger than any agent, suggesting that embedded agency can provide a natural formalization of the big world hypothesis and of continual learning.

To formalize an agent embedded in an environment, we define a *universal-local environment*: a Markov process whose transition dynamics can simulate an agent within a finite portion of its state-space. *Computational universality* guarantees that the environment can simulate any algorithm (Church, 1936; Turing, 1937). *Uniform locality* decomposes the environment's transition dynamics into a collection of identical Markov processes, each operating on a finite portion of the state-space. An embedded agent is then represented as an automaton simulated within the environment, formalized as one of these local Markov processes. When the automaton's dynamics are determined solely by its input and output, we prove it interacts with a partially observable Markov decision process: it receives inputs (observations), updates its internal state, and produces outputs (actions). We then propose *interactivity*, defined through algorithmic complexity (Kolmogorov, 1965; Solomonoff, 1964; Chaitin, 1966), to measure an automaton's ability to continually adapt its future input-output behaviour. An interactivity-seeking agent pursues behaviour that is increasingly complex while remaining predictable given its past experience. Crucially, we prove that an agent's interactivity is constrained by its capacity, which can be allocated either to increase its behavioural complexity or to improve its behavioural predictability.

Universal artificial intelligence similarly considers universal environments (Hutter, 2000, 2005), in which the uncomputable AIXI agent was extended to an embedded formulation (Orseau and Ring, 2012). However, these works did not consider the problem of learning under limited capacity. By considering a universal environment that is also local, we show that an interactivity-seeking agent must

continually adapt to its experience, regardless of its capacity. Interactivity also relates to intrinsic motivation objectives (Schmidhuber, 1991; Chentanez et al., 2004), such as forecasting complexity (Grassberger, 1986), statistical complexity (Crutchfield and Young, 1989), predictive information (Bialek et al., 2001; Still and Precup, 2012), and light cone complexity (Shalizi et al., 2004; Aaronson et al., 2014). These objectives are defined through Shannon information as measures of past-future dependence in the agent's behaviour, requiring a probability distribution over behaviour sequences. Interactivity, however, uses algorithmic information, allowing it to operate directly on individual sequences, without requiring probability distributions. Since an agent's behaviour is an individual sequence rather than a probability distribution, this formulation provides a natural computational framework for continual learning. Moreover, by tying the agent's objective to its computational capacity, which is naturally constrained relative to the environment, our formalism further captures the big world hypothesis.

Lastly, we develop a reinforcement learning algorithm for maximizing interactivity, and use it to construct a task that evaluates a learning algorithm's capability for continually adaptive behaviour. Specifically, we operationalize algorithmic complexity in terms of the prediction error. Interactivity is then measured as the reduction in this error attributable to continual learning, relative to a baseline agent that stops learning. Interactivity-seeking behaviour thus involves learning a policy to steer the agent's behaviour to new experiences that are learnable, but that would have high prediction error without learning. We show that interactivity-seeking agents create their own non-stationarity by changing their policy, thereby satisfying key desiderata of the continual learning problem: every agent is constrained by its capacity, and any agent that stops learning is suboptimal. Our results indicate that, in this setting, deep nonlinear networks struggle to sustain interactivity, whereas deep linear networks effectively scale their interactivity with increased capacity.

## 2 Background

We formalize an embedded agent from a computational perspective, viewing the environment as a computational process that also simulates the agent. Specifically, we consider a computationally universal environment that, through its state transitions, simulates the computational steps of any algorithm. Our approach is general, by making use of the Church-Turing thesis, which asserts that all computationally universal systems are equivalent in what they can simulate, and that any such system can simulate another (Church, 1936; Turing, 1937). This allows us to adopt a particular model of computation (*e.g.*, Turing machines) while retaining a general class of environments that are capable of simulating an embedded agent.

We characterize the capabilities of an agent, relative to its environment, in terms of its input-output behaviour as a finite sequence (*i.e.*, a string). In particular, we use the algorithmic complexity of a string, which is the length of the shortest program that computes it and halts (Kolmogorov, 1965; Solomonoff, 1964; Chaitin, 1966).

**Definition 1** (Algorithmic Complexity). *Given strings $x, y \in \Sigma^*$, where $\Sigma$ is a finite symbol-set and $\Sigma^*$ is the set of strings, the conditional algorithmic complexity is the length of the shortest program, $|c|$, that halts and outputs $x$ given $y$ as input,*

$$\mathbb{K}_{\mathcal{U}}(x|y) := \min\{|c| : \mathcal{U}(c, y) = x\},$$

*where $\mathcal{U}$ is a reference universal machine. The unconditional algorithmic complexity is given by $\mathbb{K}_{\mathcal{U}}(x) := \mathbb{K}_{\mathcal{U}}(x \mid \epsilon)$, where $\epsilon$ is the empty string.*

While algorithmic complexity depends on the choice of a reference universal machine, any specific choice affects the algorithmic complexity by, at most, an additive constant independent of the specific string (Li and Vitányi, 2019). Moreover, in this work, we will consider the given computationally universal environment, defineed in the next section, as the canonical reference universal machine.

## 3 A Universal-Local Environment

We begin by defining *universal-local environments*, a general class of environments in which an agent can be embedded. These environments are characterized by two key properties: computational universality (Section 3.1), which establishes that any algorithm can be simulated by the environment's transition dynamics, and uniform locality (Section 3.2), which ensures that any finite computation can be confined to a bounded portion of the environment's state-space. In Section 4, we use these properties to define an embedded agent as an automaton simulated on the environment's state-space.

## 3.1 Markov Representation of a Computationally Universal Environment

We first connect computational processes with Markov processes defined over a countable state-space with a transition function that is computable in polynomial-time with respect to the size of the state.

**Definition 2.** *An algorithmic Markov process, $\mathcal{E} = (\Omega, \Xi, \mathbb{T})$, is a discrete process defined on a countable state-space, $\Omega := \{\omega : \Xi \to \Sigma : |\omega| < \infty\}$, where $\Sigma$ is a finite symbol set with distinguished blank symbol $\square \in \Sigma$, $\Xi$ is a countable set used for indexing, and $|\omega| := |\{\xi \in \Xi : \omega(\xi) \neq \square\}|$ counts the number of non-blank symbols. Given an initial state $\omega \in \Omega$ with $|\omega| < \infty$, the process produces the next state $\omega' = \mathbb{T}(\omega)$, such that $|\omega'| < \infty$ and where the transition function, $\mathbb{T} : \Omega \to \Omega$, is computable in time $O(\texttt{poly}(|\omega|))$.*

The significance of this formulation is that any Turing machine can be represented as an algorithmic Markov process. Specifically, the Markov state represents the entire configuration of the Turing machine, including its head position, current tape contents, and control state. The Markov transition function represents the Turing machine's transition function, which is a lookup table that can be applied to the Markov state in polynomial-time. We will consider only deterministic transitions, but non-deterministic transitions are also possible, and would involve multiple possible next states.

**Proposition 1** (Representing Turing machines). *The computational process followed by a Turing machine can be represented as an algorithmic Markov process.*

All proofs of propositions and theorems can be found in Section B of the Appendix. One consequence of Proposition 1 is that there exists a universal Markov process (an algorithmic Markov process corresponding to a universal Turing machine). This result highlights how an algorithmic Markov process is more general than the Markov processes typically considered in reinforcement learning. In particular, a universal Markov process is capable of simulating any algorithm, which is crucial for defining an embedded agent in Section 4.

## 3.2 Defining Uniform Locality with Boundaried Markov Processes

Intuitively, uniform locality means that the transition function can be decomposed into identical local transition functions with dynamics that are determined by a finite portion of the state-space. To make this precise, we first formalize a finite portion of the state-space as a *substate-space*.

**Definition 3.** *Given an algorithmic Markov process, $\mathcal{E} = (\Omega, \Xi, \mathbb{T})$, a substate-space, $\Omega|_F$, is defined as a restriction of the state-space, $\Omega$, to a finite index-set, $F \in \mathcal{F}(\Xi) := \{I : I \subset \Xi, |I| < \infty\}$: $\Omega|_F = \{\omega|_F : \omega \in \Omega\}$ where $\omega|_F : F \to \Sigma$ is defined by, $\omega|_F(\xi) := \omega(\xi)$ for all $\xi \in F$.*

We now consider the transition function restricted to a substate-space, $\Omega|_F$. In particular, we define a boundaried Markov process in which the restricted transition function, $\mathbb{T}|_F$, depends on both the substate-space, $\Omega|_F$, and an additional substate-space, $\Omega|_{b(F)}$, called the boundary-space.

**Definition 4.** *Given an algorithmic Markov process, $\mathcal{E} = (\Omega, \Xi, \mathbb{T})$, a substate-space, $\Omega|_F$, admits a $k$-horizon boundaried Markov process if there exists a substate-space $\Omega|_{b^k(F)}$ (referred to as a boundary-space) and a restricted transition function $\mathbb{T}|_F^k : \Omega|_F \times \Omega|_{b^k(F)} \to \Omega|_F$ that is equivalent to the $k$-step transition function on the substate-space: $\mathbb{T}|_F^k(\omega|_F, \omega|_{b^k(F)}) = \mathbb{T}^{(k)}(\omega)|_F$ for all $\omega \in \Omega$. We denote the $k$-horizon boundaried Markov process as $\mathcal{E}|_F^k = (\Omega|_F, \Omega|_{b^k(F)}, \mathbb{T}|_F^k)$, and, if $k = 1$, we refer to it simply as a boundaried Markov process, $\mathcal{E}|_F = (\Omega|_F, \Omega|_{b(F)}, \mathbb{T}|_F)$.*

Typically, the size of the boundary-space increases as the transition horizon $k$ becomes larger. This is because the current substate, $\omega|_F \in \Omega|_F$, and the current boundary, $\omega|_{b(F)} \in \Omega|_{b(F)}$, only determine the next substate, $\omega'|_F \in \Omega|_F$, and not the next boundary.

An algorithmic Markov process is uniformly local if every index admits a boundaried Markov process that is isomorphic to a single reference boundaried Markov process.

**Definition 5** (Uniform Locality). *An algorithmic Markov process, $\mathcal{E} = (\Omega, \Xi, \mathbb{T})$, is uniformly local if there exists a boundary function, $b : \Xi \to \mathcal{F}(\Xi)$ and a reference $\xi_\star \in \Xi$, such that every $\xi \in \Xi$ admits a boundaried Markov process isomorphic to the reference $\xi_\star$. That is, there exist bijections $\alpha_\xi : \Omega|_{\{\xi\}} \to \Omega|_{\{\xi_\star\}}$ and $\beta_\xi : \Omega|_{b(\{\xi\})} \to \Omega|_{b(\{\xi_\star\})}$ satisfying, for all $\omega \in \Omega$,*

$$\alpha_\xi\left(\mathbb{T}|_{\{\xi\}}(\omega|_{\{\xi\}}, \omega|_{b(\{\xi\})})\right) = \mathbb{T}|_{\{\xi_\star\}}(\alpha_\xi(\omega|_{\{\xi\}}), \beta_\xi(\omega|_{b(\{\xi\})})).$$

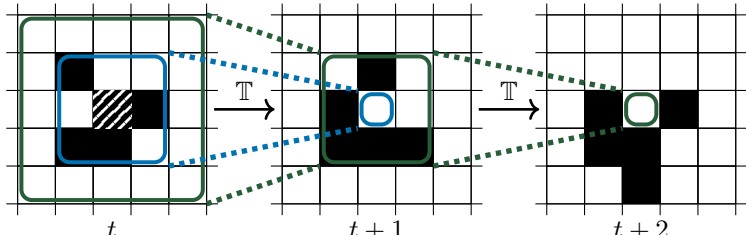

Figure 2: **Conway's Game of Life is a cellular automaton and an example of a universal-local environment.** The state-space is an infinite 2D grid, $\Xi = \mathbb{Z}^2$, where each cell in the grid takes one of two values, $\Sigma = \{\text{black}, \text{white}\}$. Every cell uses the same local transition function in which the boundary-space consists of its 8 adjacent neighbours. The blue and green borders (*left*) correspond to 1 and 2 horizon boundary-spaces, which determine the center cell at time-steps $t + 1$ (*middle*) and $t + 2$ (*right*), respectively. Longer-term transition dynamics depend on larger boundary-spaces.

Uniform locality guarantees that the transition function can be computed by simultaneously applying an identical local transition function to each singleton of the state-space. Consequently, the transition dynamics for any substate-space $\Omega|_F$ can be determined from the collective boundary, defined by $b(F) = \bigcup_{\xi \in F} b(\{\xi\})$. This ensures that every substate-space admits a boundaried Markov process.

Thus, we use the term *universal-local environment* for a universal Markov process that is also uniformly local. We call it an environment because, as we will show, an embedded agent can be simulated as a boundaried Markov process within it.

### 3.3 Example of a Universal-Local Environment: Conway's Game of Life

Conway's Game of Life (or Life) is an example of a universal-local environment (Conway, 1970). This environment is computationally universal because it can simulate a universal Turing machine (Berlekamp et al., 1982; Rendell, 2011). A substate-space in Life is a finite subset of cells on the grid and the possible values taken by those cells. Furthermore, Life is uniformly local because the one-step transition dynamics for each individual cell is determined by its 8 adjacent neighbouring cells, which defines the boundary-space (see Figure 2).

We point out Life as an existence proof for universal-local environments, but we are not suggesting to program an agent within it. Instead, we use our formalism to characterize the constraints that would be faced by an agent if it were embedded in any such environment.

## 4 A Computationally-Embedded Agent

We represent an embedded agent as an automaton simulated within the universal-local environment, formalized as a boundaried Markov process. Specifically, we prove that when the automaton's boundary space coincides with its input-output space, it constitutes a formal agent-environment boundary (Jiang, 2019; Harutyunyan, 2020). Consequently, the automaton is equivalent to a stateful policy interacting with a partially observable Markov decision process. We then introduce *interactivity* to measure a capability for continually adaptive behaviour, proving that interactivity-seeking automata are constrained by their computational capacity.

### 4.1 Embedding an Automaton in a Universal-Local Environment

A universal-local environment can simulate any algorithm; this property enables us to define an automaton, $\mathcal{A}$, within the environment's state space, $\Omega$, such that its operation is simulated by the environment's transition dynamics, $\mathbb{T}$. Moreover, uniform locality ensures that the automaton can be represented as a boundaried Markov process (see Figure 3, left).

**Definition 6.** *Given a universal-local environment, $\mathcal{E} = (\Omega, \Xi, \mathbb{T})$, an embedded automaton is defined by $\mathcal{A} := (\Omega|_X, \Omega|_Y, \Omega|_\Theta, u, \pi)$, where $\Omega|_\Theta$ is the automaton's internal state space, $\Omega|_X, \Omega|_Y$ are input and output spaces, $\pi : \Omega|_X \times \Omega|_\Theta \to \Omega|_Y$ is the output function, and $u : \Omega|_X \times \Omega|_\Theta \to \Omega|_\Theta$ is the automaton's internal state update function.*

Relating this to an agent in reinforcement learning, we may think of the inputs as observations, the internal state as the parameters of a function approximator, the outputs as actions, the internal state

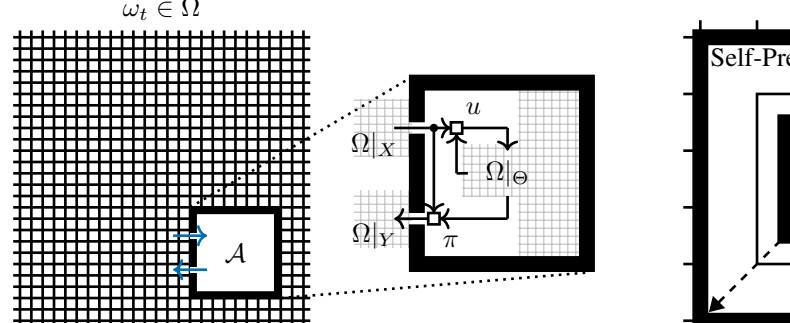

Figure 3: **An illustrative depiction of a computationally-embedded agent interacting with its environment.** An embedded automaton, $\mathcal{A} = (\Omega|_X, \Omega|_Y, \Omega|_\Theta, u, \pi)$, represents an agent embedded within the universal-local environment. **Left:** The agent is simulated by a boundaried Markov process within the environment, in which it iteratively receives an input from the environment, $x \in \Omega|_X$, produces the corresponding output $y = \pi(x; \theta) \in \Omega|_Y$, and updates its internal state, $\theta' = u(x; \theta) \in \Omega|_\Theta$. **Right:** We will also consider an idealized setting in which a self-predicting agent exerts full control over its experience by reading and writing to an internal boundary-space, $\Omega|_{b(\Theta)}$.

update function as a learning rule, and the output function as a policy. The input-space may also provide an external reward to the automaton, but this need not be the case.

**Proposition 2** (Automaton-Environment Relationship). *Given an embedded automaton, $\mathcal{A} = (\Omega|_X, \Omega|_Y, \Omega|_\Theta, u, \pi)$, if there exists a horizon $k$ such that $b^k(\Theta) = X$, then:*

1. *The automaton is equivalent to a $k$-horizon boundaried Markov process*

2. *The automaton's environment is a partially observable Markov decision process.*

3. *The automaton's interaction with the environment is equivalent to a stateful policy.*

Now that we have defined both the embedded automaton and its environment within the same universal-local environment, we can show that any such automaton is constrained by the size of its internal state-space, which determines its memory and computational capacity.

**Proposition 3** (Implicitly Constrained). *The capacity of an embedded automaton is upper bounded by the size of its internal state space, $|\Theta|$, which is finite. Thus, there exists input-output sequences that the automaton cannot realize.*

An embedded automaton is constrained by its capacity, which constrains the behaviours that it can produce. While more complex behaviour can be produced with more capacity, we will show that this constraint specifically limits its ability to produce behaviour that continually adapts to past experience.

### 4.2 Interactivity as a Computational Measure of Adapatability

An agent's capability for learning can be characterized by its ability to adapt its future behaviour using its past experience. We propose *interactivity* to measure this capability directly in terms of algorithmic complexity. Specifically, interactivity measures the difference between the algorithmic complexity of future behaviour with and without conditioning on past experience.

Following Proposition 2, we represent an embedded agent as an embedded automaton $\mathcal{A}$ where its input space constitutes its boundary-space, $\Omega|_X = \Omega|_{b^k(\Theta)}$. Thus, the behaviour of the agent is determined by the values taken on the boundary-space, $b_t \in \Omega|_{b^k(\Theta)}$ which combines the input and the corresponding output, $b_t = (x_t, y_t)$. At time $t$, the behaviour can be separated into finite sequences: the past, $b_{0:t-1} := b_0 b_1 \cdots b_{t-1}$, and the $T$-horizon future, $b_{t:t+T-1} := b_t b_{t+1} \cdots b_{t+T-1}$.

**Definition 7** (Interactivity). *Given an embedded automaton, $\mathcal{A} := (\Omega|_X, \Omega|_Y, \Omega|_\Theta, u, \pi)$, we define its $T$-horizon interactivity relative to its environment, $\mathcal{E}$, as the difference between the algorithmic complexity of its future behaviour with and without conditioning on its past behaviour,*

$$\mathbb{I}_T(\mathcal{A} \,|\, x_t, b_{0:t-1}) := \mathbb{K}_\mathcal{E}(b_{t:t+T-1} \,|\, \epsilon) - \mathbb{K}_\mathcal{E}(b_{t:t+T-1} \,|\, b_{0:t-1}),$$

*where $\epsilon$ is the empty string, $x_t$ is the current observation, $\theta_{t-1}$ is the internal state, $y_t = \pi(x_t; \theta_{t-1})$ is the action, $\theta_t = u(x_t; \theta_{t-1})$ is the next internal state, and $b_t = (x_t, y_t)$ is the next behaviour tuple.*

That is, interactivity measures the predictable complexity of an agent's future behaviour, given its past behaviour. Interactivity is high if both (i) the future behaviour, $b_{t:t+T-1}$, has high unconditional algorithmic complexity, and (ii) the past behaviour, $b_{0:t-1}$, is predictive of this future behaviour, thereby yielding a low conditional algorithmic complexity. Interactivity-seeking behaviour thus balances complexity and predictability with respect to the agent's past behaviour, similar to how definitions of open-endedness balance novelty and learnability with respect to an observer (Hughes et al., 2024).

### 4.3 An Interactivity-Seeking Agent Faces a Big World

The interactivity of any embedded agent is always constrained by its capacity. That is, an interactivity-seeking agent can only sustain a given level of interactivity with a given capacity. However, an interactivity-seeking agent can always use additional capacity to achieve higher interactivity.

**Theorem 1.** *Given an embedded agent, $\mathcal{A} = (\Omega|_X, \Omega|_Y, \Omega|_\Theta, u, \pi)$, its maximum interactivity is asymptotically upper and lower bounded by a quantity that depends on its capacity.*

The goal of an interactivity-seeking agent can be understood as balancing the complexity and predictability of its future behaviour, given its past behaviour. Seeking to achieve this goal requires that the agent continually adapt to its experience, learning about its computational limitations, and tracking its environment with predictions learned within these limitations. This suggests the following interactivity thesis:

*Interactivity measures a capability for continually adaptive behaviour.*

We refer to this as the interactivity thesis, rather than a hypothesis, to reflect its speculative and philosophical nature. With low interactivity, the future behaviour of an agent is either i) simple, or ii) complex and unpredictable. In either case, the thesis asserts that the agent's capability for continually adaptive behaviour is limited. A simple agent has a limited range of possible behaviours and thus has a relatively lower capability for adaptive behaviour. A complex agent could have a relatively high capability for adaptive behaviour, but only if its future behaviour is influenced by, and can be predicted from, past experience. Embracing the interactivity thesis naturally leads to a spectrum of adaptive capability.

## 5 Maximizing Agent-Relative Interactivity with Reinforcement Learning

Maximizing interactivity poses a fundamental computational challenge: it depends on algorithmic complexity, which is generally uncomputable. While algorithmic complexity can be computed for finite automata through exhaustive program enumeration (Li and Vitányi, 2019), this requires computational resources that exceed the automaton's own capacity. Consequently, an interactivity-seeking agent must approximate interactivity rather than compute it exactly.

To approximate interactivity, we take a distortion-rate view of algorithmic complexity, which measures complexity in terms of prediction error under a constrained reference machine, rather than under an unconstrained universal Turing machine (Vereshchagin and Vitányi, 2010). The embedded agent, being an automaton, provides a natural choice for the constrained reference machine. We then augment this agent to include a predictor function as a function of agent-state, and measure agent-relative complexity by the incurred temporal difference errors under this predictor.

**Definition 8.** *Given an embedded agent $\mathcal{A} = (\Omega|_X, \Omega|_Y, \Omega|_\Theta, u, \pi)$ with a predictor $v : \Omega|_{X \cup Y} \times \Omega|_\Theta \to \Omega|_{X \cup Y}$ and $\gamma \in [0, 1]$, the agent-relative complexity of future behaviour $b_{t:t+T-1}$, conditioned on past behaviour $b_{0:t-1}$, is the sum of future temporal difference errors,*

$$\hat{\mathbb{K}}_{\mathcal{A}}(b_{t:t+T-1}|b_{0:t-1}) := \sum_{k=0}^{T-1} \left( b_{t+k} + \gamma v(b_{t+k}; \theta_{t+k-1}) - v(b_{t+k-1}; \theta_{t+k-1}) \right)^2,$$

*where $\theta_{t+k-1}$ is the agent-state after processing $b_{0:t-1}$ followed by $b_{t:t+k-1}$ through repeated application of the agent's update function $u$.*

Complexity, measured in this way, is relative to the agent's capabilities. If the future prediction errors are large (small), then the future behaviour is relatively complex (simple). These future prediction errors depend crucially on the agent through its predictor, its current agent-state, its

learning algorithm, its policy, and its observations from the environment. With this prediction error formulation of complexity, we now consider the agent-relative interactivity,

$$\hat{\mathbb{I}}_T(\mathcal{A} \,|\, x_t, b_{0:t-1}) := \hat{\mathbb{K}}_{\mathcal{A}}(b_{t:t+T-1} \,|\, \epsilon) - \hat{\mathbb{K}}_{\mathcal{A}}(b_{t:t+T-1} | b_{0:t-1}).$$

## 5.1 Learning to Maximize Agent-Relative Interactivity

We now develop a reinforcement learning algorithm for maximizing agent-relative interactivity. Our algorithm is summarized in the following three steps: (i) learning a prediction of the discounted future behaviour using a value function, (ii) computing the agent-relative interactivity, defined as the difference between static prediction errors (unconditional complexity) and dynamic prediction errors (conditional complexity), and (iii) meta-learning a policy to maximize agent-relative interactivity.

The first step involves learning a prediction of the future input-output behaviour, where we consider the deterministic setting and omit the expectation over trajectories. This value function predicts both future observations and actions, similar to successor features (Barreto et al., 2017) and the successor representation (Dayan, 1993). Specifically, with temporal difference learning (Sutton, 1984, 1988), we train a value function to predict the discounted sum of future input-output behaviour,

$$v(b_t; \theta_t) \approx \sum_{k=0}^{\infty} \gamma^k b_{t+k+1}, \quad \delta_{t+k}(\theta) = b_{t+k} + \gamma v(b_{t+k}; \theta) - v(b_{t+k-1}; \theta),$$

where $\theta_{t+k}$ can be understood as dynamic parameters which are updated using semi-gradient TD(0): $\theta_{t+k} = \theta_{t+k-1} + \eta \delta_{t+k}(\theta_{t+k-1}) \nabla_\theta v(b_{t+k-1}; \theta)\big|_{\theta=\theta_{t+k-1}}$, where $\eta$ is a step-size.

For the second step, we measure the conditional and unconditional complexity terms using dynamic and static temporal difference errors, respectively. From the first step, we can readily compute the conditional complexity using the dynamic temporal difference errors just described,

$$\hat{\mathbb{K}}_{\mathcal{A}}(b_{t+1:t+T} | b_{0:t-1}) = \sum_{k=0}^{T-1} \delta_{t+k}^2(\theta_{t+k-1}).$$

Unlike the conditional complexity term, the unconditional complexity term is not conditioned on past experience, which means means that it cannot be computed using the dynamic parameters. Instead, computing the unconditional complexity involves a static and unchanging reference state, $\theta_{ref}$. One convenient choice is the current state, $\theta_{ref} := \theta_{t-1}$, which we adopt to yield the following form for agent-relative interactivity,

$$\hat{\mathbb{I}}_T(\mathcal{A} \,|\, x_t, b_{0:t-1}) \approx \sum_{k=1}^{T-1} \left( \underbrace{\delta_{t+k}^2(\theta_{t-1})}_{\text{static}} - \underbrace{\delta_{t+k}^2(\theta_{t+k-1})}_{\text{dynamic}} \right).$$

Lastly, we outline how a policy can be trained to maximize interactivity. To obtain the future prediction errors, we assume access to a differentiable model that we can use to roll-out a sequence of observations and actions using the current policy. Interactivity is estimated on the roll-out by computing the cumulative difference between the static and dynamic prediction errors. The policy is then updated using a gradient-based optimizer with the objective of maximizing the estimated interactivity.[1] Crucially, both the policy and value function must be continually updated to sustain interactivity: if the value function stops changing then interactivity is trivially zero, and if the policy stops changing then the value function can converge to a fixed point.

## 5.2 Maximizing Interactivity as a Continual Learning Problem

We now show that an interactivity-seeking reinforcement learning agent faces a continual learning problem. Intuitively, this is because interactivity-seeking behaviour is directed towards experience which can be predicted by a dynamic value function that learns from experience, but which cannot be predicted by the current value function if it were to remain static. Specifically, interactivity-seeking agents are suboptimal if they stop learning, such as when the parameters of their policy and value stop changing. Similar to our previous result on embedded automata (Theorem 1), we also show that the agent's maximum possible interactivity increases as the its capacity increases.

---

[1]Optimizing the dynamic errors involves meta-gradients through the value function's parameter update.

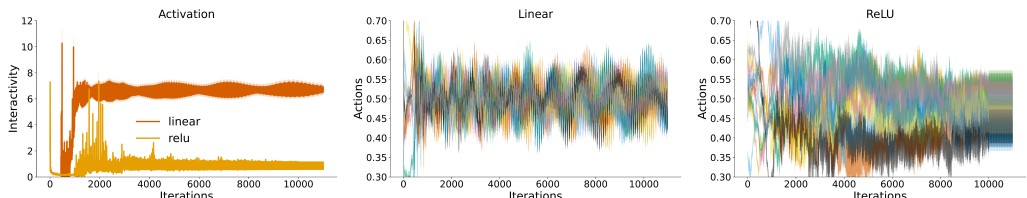

Figure 4: **Deep `ReLU` networks fail to sustain performance on the interactivity evaluation task.** Interactivity-seeking simulates a big world to evaluatesa learning algorithm's capability for continually adaptive behaviour. **Right:** The deep linear network sustains interactivity, whereas the deep `ReLU` collapses in performance. **Middle:** Each colour corresponds to one component of the action vector. The deep linear policy learns to produce actions following a non-stationary wave, which can be predicted by a dynamic linear function. **Left:** The deep `ReLU` policy fails to produce actions with any predictable structure, despite its increased expressivity

**Theorem 2** (Big World). *An agent that seeks to maximize its agent-relative interactivity is (i) limited by its finite capacity and, (ii) suboptimal if it stops learning.*

The desiderata of Theorem 2 were previously described as conditions for a big world simulator (Kumar et al., 2024). Thus, interactivity-seeking appears to be a general problem setting which captures the big world hypothesis: the best use of an agent's limited capacity is to continually adapt.

## 6 Evaluating Continual Adaptation With Agent-Relative Interactivity

We now use interactivity-seeking to construct a synthetic problem to evaluate a learning algorithm's capability for continually adaptive behaviour. In particular, we consider a setting in which the agent only observes its own actions (see the self-predicting agent in Figure 3, right). Even though such an agent has full control over its experience stream, its interactivity is still implicitly constrained by its capacity. That is, the interactivity objective depends on the parameters and learning algorithm of the value function, which the agent does not directly observe. An advantage of this evaluation approach is that it is environment-free, allowing direct evaluation of an algorithm outside of an environment, or any collected data. Instead, algorithms are directly evaluated by continually learning from their own online experience in a manner similar to self-play.

We instantiate the reinforcement learning agent outlined in Section 5 with a linear parameterization of the value function, $v(b_t; \mathbf{W}_t) := \mathbf{W}_t b_t \approx \sum_{k=0}^{\infty} \gamma^k b_{t+k+1}$. Linearity provides stability for learning online with TD(0), compared to temporal difference learning with deep nonlinear networks. For the policy parameterization, we consider a deep network architecture (using either linear or `ReLU` activations), where we normalize the output to ensure that the output has bounded range, $b_{t+1} := \text{RMSNorm}\left(\pi(b_t; \theta_t)\right)$. This policy is optimized using the model-based approach described in Section 5. Optimizing the agent-relative interactivity is a bi-level optimization problem as the dynamic prediction errors depend on the value function's learning process. Thus, our model-based approach to maximizing interactivity is similar to the cross-prop algorithm (Veeriah et al., 2017), which is an online version of model-agnostic meta-learning (Finn et al., 2017). For both the policy and the value function, we found RMSProp (Hinton et al., 2012) to balance performance and stability better than either Adam (Kingma and Ba, 2015), or vanilla gradient descent.

Our results demonstrate that a deep nonlinear policy is unable to sustain interactivity (see Figure 4, right). That is, the deep nonlinear policy is unable to plan an action sequence for which the dynamic value function has low prediction error, but for which the current static value function has high prediction error. However, we find that a deep linear policy is able to sustain interactivity. This finding suggests that interactivity-seeking agents produce non-stationarity that can lead to apparent loss of plasticity, which linear methods have been shown to avoid (Lewandowski et al., 2025b; Dohare et al., 2024). Observing the actions chosen by each policy, we found that the deep linear policy learned to produce actions with predictable structure, resembling a non-stationary wave (see Figure 4, middle). These actions can be locally predicted by a linear function, but global prediction requires a dynamic linear function. In contrast, the deep nonlinear policy learned failed to produce actions with any predictable structure (see Figure 4, left). Furthermore, in Figure 5, we found that deep linear networks are also capable of increasing their interactivity with more capacity, in the form of deeper or wider networks. These findings demonstrate that interactivity-seeking simulates the challenges of a big world in which any policy is limited by its finite capacity, and suboptimal if it stops learning.

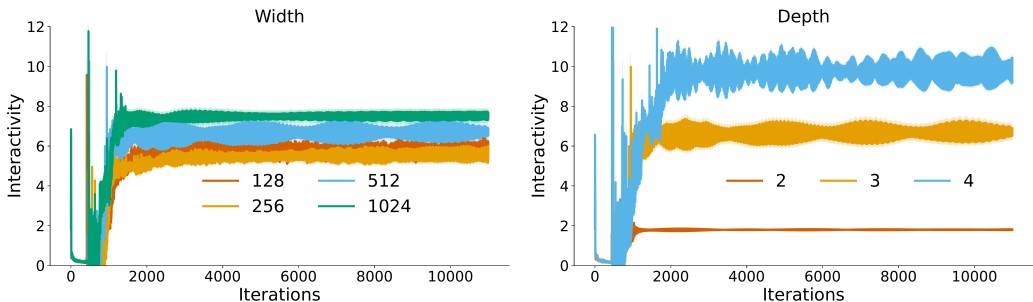

Figure 5: **Deep linear networks are capable of sustaining higher interactivity with more computational resources.** By increasing the width and depth of the deep linear network, we increase the network's capacity for continual adaptation, which allows it more quickly change its linear function approximator. **Left:** Increasing width marginally increases the sustained interactivity. **Right:** Increasing depth results in a large increase in interactivity, as well as more oscillatory behaviour.

## 7 Discussion

In this paper, we introduced a computationally-embedded perspective on the big world hypothesis, where we (i) characterize the implicit constraint faced by an embedded agent, (ii) propose interactivity as a computational measure of adaptability, and (iii) develop a reinforcement learning algorithm for maximizing interactivity. We show that interactivity-seeking leads to the common desideratum of the continual learning problem in which any agent that stops learning is suboptimal.

Our work departs from dogmas common in reinforcement learning (Abel et al., 2024) by formalizing the agent as embedded in the environment, and by proposing interactivity as an objective defined relative to an agent and its learning algorithm. Interactivity-seeking agents are implicitly constrained because the interactivity objective depends on components, such as parameters, that are not directly observable but evolve through learning. As we demonstrated in the environment-free evaluation task, this perspective has a critical consequence: an interactivity-seeking agent must continually adapt rather than converging to a fixed point. Unlike existing continual learning problems that often involve manually designed non-stationarity, an interactivity-seeking agent autonomously produces the non-stationarity through its learning algorithm. This task thus provides several unique challenges: (i) the task complexity scales with the complexity of the learning algorithm, (ii) hyperparameters influence the non-stationarity produced by the learning algorithm, requiring more careful procedures for tuning (Mesbahi et al., 2025), and (iii) the task incentivizes the design of continual learning algorithms that can both sustain plasticity to learn from new experience (Lyle et al., 2024; Dohare et al., 2024), and use plasticity and prior learning to create more new experience through non-stationarity.

Beyond serving as an evaluation task for continual learning, interactivity can also serve as an auxiliary objective for agents in reinforcement learning environments. Unlike the environment-free evaluation task investigated in this paper, estimating the interactivity of an agent in an environment would involve learning a model of the environment to calculate meta-gradients, or learning a value function that directly approximates interactivity. The change in temporal difference error due to learning would then serve as an intrinsic reward, similar to curiosity-driven methods, to incentivize directed exploration.

This practical potential for directed exploration suggests a refinement of the interactivity thesis. We speculate that if an agent can sustain a particular level of interactivity, then it possesses the capacity to learn any behaviour with equal or lower interactivity—including those that maximize the discounted cumulative sum of a reward signal. That is, the ability to sustain high interactivity may reflect a more general capacity for acquiring goal-directed behaviours.

### Acknowledgments

We would like to thank Saurabh Kumar and Hong Jun Jeon for helpful discussion during earlier stages of this work, David Abel and Anna Harutyunyan for their encouraging discussion regarding dogmas in reinforcement learning, and the anonymous reviewers for their feedback. The research is supported in part by the Natural Sciences and Engineering Research Council of Canada (NSERC), the Canada CIFAR AI Chair Program, and the Digital Research Alliance of Canada.

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

# A   Experimental Details for Behavioural Self-Prediction

The code for the experiments can be found at: https://github.com/AlexLewandowski/bigger-world-interactivity

The problem that we consider involves a value function predicting a policy's actions, where the policy is trained to maximize interactivity. Specifically, the policy is trained to produce actions which have large prediction errors for the current static value function but which have low prediction errors for a dynamic value function that is continually learned. We consider a policy that maps its previous action to a new action, meaning that there is no external environment providing observations.

The value function, which we restrict to be linear, is tasked with predicting the future behaviour of the policy iteratively and online. At the first timestep, we randomly initialize the parameters of the policy $\theta_0 \sim p(\theta)$ and of the value function, $\mathbf{W}_0 \sim p(\mathbf{W})$, using standard distributions for neural network initialization. We then randomly sample the initial action, $b_0 \sim N(0, 1/d)$, where $d = 1000$ denotes the dimensionality of the action, $b \in \mathbb{R}^d$. The policy, $\pi_\theta$, is then trained to maximize its interactivity whereas the value function $v_{\mathbf{W}}$ is trained to predict the policy using TD(0).

The following steps are repeated at each timestep $t$,

- A trajectory of $T$ actions is produced by the policy, $\pi_{\theta_t}$.
- A copy of the current value function is frozen, and we record the temporal difference errors incurred along this trajectory: $\{\delta_{t+1}(\mathbf{W}_{t+1}, \theta_t), \ldots, \delta_{t+T}(\mathbf{W}_{t+T}, \theta_t)\}$. The sum of squared temporal difference errors provides a measure of the unconditional algorithmic complexity, $\hat{\mathbb{K}}_{\mathcal{A}}(b_{t:T}|\epsilon) = \sum_{k=1}^{T} \delta_{t+k}^2(\mathbf{W}_t, \theta_t)$.
- A copy of the current value function is updated dynamically along the trajectory, and we record the temporal difference errors incurred along this trajectory: $\{\delta_{t+1}(\mathbf{W}_{t+1}, \theta_t), \ldots, \delta_{t+T}(\mathbf{W}_{t+T}^-, \theta)\}$. Where each $\mathbf{W}_{t+k}^-$ corresponds to the updated parameters of the value function. The sum of squared temporal difference errors provides a measure of the conditional algorithmic complexity, $\hat{\mathbb{K}}_{\mathcal{A}}(b_{t:t+T-1}|b_{0:t-1}) = \sum_{k=1}^{T} \delta_{t+k}^2(\mathbf{W}_{t+k}^-, \theta_t)$.
- We update the policy using a single step from a gradient-based optimizer, maximizing interactivity, but written here as minimizing negative interactivity,

$$J(\theta) = -\sum_{k=1}^{T} \delta_{t+k}^2(\mathbf{W}_t, \theta) - \delta_{t+k}^2(\mathbf{W}_{t+k}^-, \theta).$$

- We take an action using the updated policy, $b_{t+1} = \pi(b_t; \theta_{t+1})$.
- We now update the value function with a single step of TD(0) using the temporal difference error, $\delta_t = b_{t+1} + \gamma v(b_{t+1}, \mathbf{W}_t) - v(b_t, \mathbf{W}_t)$.

## A.1   Interpreting Interactivity Maximization As A Continual Learning Benchmark

Our theoretical and experimental results show that maximizing interactivity requires both fast adaptation to increase complexity with higher prediction errors and stability to sustain adaptation over time. That is, maximizing interactivity involves the canonical plasticity-stability trade-off of continual learning (Grossberg and Grossberg, 1982; Parisi et al., 2019). Our empirical results demonstrate that deep nonlinear baselines fail at striking this balance, whereas deep linear networks appear to naturally achieve this balance. This suggests that this synthetic benchmark isolates the key challenge in continual learning, while also not requiring outside data or environments. This is significant because few environments are designed specifically to evaluate continual adaptation. Thus, it is suboptimal to stop learning in this setting, regardless of the capacity of the algorithm or function approximator.

## A.2   Limitations of Experiments

Our experiments used relatively shallow networks, with a maximum depth of $D = 4$. However, with the meta-gradient calculation over a finite horizon of $T = 10$, the effective depth of the networks

during auto-differentiation is $T \cdot D = 40$. Meta-gradient methods for deep networks can exhibit more pathological learning dynamics due to increased curvature, leading to instability that could partially explain the discrepancy between linear and nonlinear networks. Understanding how to control curvature using only first-order methods is key for effective meta-gradient descent in this setting.

The meta-gradient method poses several limitations in scaling. Ideally, we would prefer to scale both the horizon and the capacity of the function approximator. However, because meta-gradients involve a second-order term (the hessian) and because the effective depth is multiplicative in the horizon and the depth of the network, this makes simultaneously scaling both costly. A more promising direction involves bootstrapping meta-gradients (Flennerhag et al., 2022), other first-order approximation (Nichol et al., 2018), or more effective regularizers (Lewandowski et al., 2025a).

Experimental evaluation in this setting also requires special consideration. Holding the agent fixed for evaluation, as is commonly done in machine learning, is not appropriate given that interactivity is defined as an online objective. In addition, standard approaches to hyperparameter tuning may not be feasible for evaluating the long-term performance of a continual learning agent (Mesbahi et al., 2025). Overcoming these obstacles to more fairly assess dependence on hyperparameters would require re-evaluation of several components of empirical practice in machine learning.

## B   Proofs

*Proof of Proposition 1.* Let $M = (Q, \Sigma', \Gamma, \delta, q_0, \square', F)$ be a Turing machine where:

- $Q$ is the finite set of states

- $\Sigma'$ is the input alphabet

- $\Gamma$ is the tape alphabet with $\Sigma' \subset \Gamma$

- $\delta : Q \times \Gamma \to Q \times \Gamma \times \{L, R\}$ is the transition function

- $q_0 \in Q$ is the initial state

- $\square' \in \Gamma$ is the blank symbol

- $F \subseteq Q$ is the set of final states

Our proof constructs an algorithmic Markov process, $\mathcal{E} = (\Omega, \Xi, \mathbb{T})$. Specifically, it involves showing how the Turing machine can be represented on a state-space $\Omega$, with its transition function represented in the Markov transition function, $\mathbb{T}$.

### Constructing the state-space ($\Omega$)

Let $\Xi = \mathbb{Z}$ denote the integers for tape positions, $\Sigma = ((Q \cup \{\square''\}) \times \Gamma) \cup \{\square\}$ where $\square$ is the Markov blank symbol. Then each $\omega \in \Omega$ represents a complete Turing machine configuration by encoding the tape contents at each position, the current state and head position.

Specifically, for a TM configuration with state $q$, head at position $h$, and tape contents $...a_{-1}a_0a_1...$, we define:

$$\omega(\xi) = \begin{cases} (q, a_h) & \text{if } \xi = h \text{ (head position)} \\ (\square'', a_\xi) & \text{if } \xi \neq h \text{ and } a_\xi \neq \square' \\ \square & \text{otherwise} \end{cases}$$

### Constructing the transition function ($\mathbb{T}$)

The Markov transition $\mathbb{T}(\omega)$ simulates one step of the Turing machine. It first finds the head position $h$ where $\omega(h) = (q, a)$ for some $q \in Q, a \in \Gamma$. It then applies the Turing machine transition: $\delta(q, a) = (q', a', d)$ where $d \in \{L, R\}$. Lastly, we construct $\omega'$ by:

- Setting $\omega'(h) = (\square'', a')$ (write new symbol)

- Setting $\omega'(h + \text{offset}(d)) = (q', b)$ where

$$b = \begin{cases} c & \text{if } \omega(h + \text{offset}(d)) = (\square'', c) \text{ for some } c \in \Gamma \\ \square' & \text{if } \omega(h + \text{offset}(d)) = \square \end{cases}$$

and $\text{offset}(L) = -1, \text{offset}(R) = 1$

- Leaving all other positions unchanged: $\omega'(\xi) = \omega(\xi)$ for all $\xi \notin \{h, h + \text{offset}(d)\}$

At initialization, we have that the initial tape contents is finite and thus $|\omega'| < \infty$. At each step, the state remains finite, $|\omega'| < \infty$, because each step changes at most 2 positions. Note also that $\mathbb{T}$ is computable in $O(\text{poly}(|\omega|))$; finding the head and updating positions requires linear scan and constant updates. $\square$

*Proof of Proposition 2.* We prove each part in sequence.

**Part 1:** The automaton is equivalent to a $k$-horizon boundaried Markov process.

Given that $b^k(\Theta) = X$, the $k$-step transition dynamics on the internal state space $\Omega|_\Theta$ depends only on the current internal state and the current observation.

We define two $k$-horizon boundaried Markov process. First, for the internal statel, we have: $\mathcal{E}|_\Theta = (\Omega|_\Theta, \Omega|_X, \mathbb{T}|_\Theta^k)$ where $\mathbb{T}|_\Theta^k(\theta, x)) := u(x; \theta)$. Next, for the policy, we have: $\mathcal{E}|_Y = (\Omega|_Y, \Omega|_{X \cup \Theta}, \mathbb{T}|_Y^k)$ where $\mathbb{T}|_Y^k(y, \theta, x) := \pi(x; \theta)$.

That is, both the policy, $\pi$, and the internal state update function, $u$, is simulated by the $k$-step composition of the environment's transition function. By the definition of uniform locality and the condition $b^k(\Theta) = X$, the $k$-step transition on $\Omega|_\Theta$ is fully determined by the current internal state $\theta \in \Omega|_\Theta$ and the boundary $x \in \Omega|_X$. This establishes the equivalence.

**Part 2:** The automaton's environment is a partially observable Markov decision process.

For the automaton specifically, the environment consists of:

- **State space:** $\Omega$

- **Action space:** The output space $\Omega|_Y$

- **Observation space:** The input space $\Omega|_X$

- **Transition function:** Given the current environment state, the automaton's action $y \in \Omega|_Y$, the next environment state follows from $\mathbb{T}$

- **Observation function:** The automaton observes $x \in \Omega|_X$ from the current environment state

Since the automaton only observes $\Omega|_X$ and not the full environment state $\Omega$, this constitutes partial observability. The Markov property holds for the underlying environment state transitions via $\mathbb{T}$.

**Part 3:** The automaton's interaction is equivalent to a stateful policy acting on the environment.

The automaton maintains an internal state $\theta \in \Omega|_\Theta$ and produces actions via the output function $\pi : \Omega|_X \times \Omega|_\Theta \to \Omega|_Y$. This defines a stateful policy, $\pi(x; \theta)$, where the internal state $\theta$ is updated according to:
$$\theta_{t+1} = u(x_t; \theta_{t-1}).$$

This is precisely the definition of a stateful policy in partially observable environments, where the policy maintains internal memory (the substate $\theta$) and conditions its actions on both observations and this internal state.

$\square$

*Proof of Proposition 3.* An embedded automaton has a finite number of states, which is upper bounded by the size of its internal state space, $|\Theta|$. This upper bounds the number of unique

configurations that the internal state space can take, which thus upper bounds the capacity of the automaton.

Thus, the automaton's capacity is constrained to be upper bounded by a number of states that is less than $|\Theta|$, which is finite.

Forthermore, an embedded automaton is only capable of a limited form of computation relative to the partially observable Markov decision process. An embedded automaton is equivalent to a finite-state machine. This means that the automaton is only capable of recognizing a regular language. The partially observable Markov decision process that it faces, however, is computationally universal and equivalent in power to a Turing machine (by Proposition 1). This means that the environment can, in general, generate a recursively-enumerable language. The embedded automaton is thus implicitly computationally constrained, because of the separation between finite-state automata and the turing-complete environment in the Chomsky hierarchy (Chomsky, 1959).

Thus, for a sufficiently long behaviour sequence, the automaton must eventually return to a previous state (*i.e.*, $T >> |\Theta|$) by the pigeonhole principle. Any behavioural sequence with a period longer than $|\Theta|$ cannot be represented by the automaton.

Thus, an embedded automaton simulated in a universal-local environment is implicitly constrained: there exist input-output behaviours that it cannot realize.

See also: C.1.

$\square$

*Proof of Theorem 1.* We denote the capacity of the automaton as $C(\mathcal{A})$, and denote the maximum achievable $T$-horizon interactivity for a given automaton as,

$$\max_{\mathcal{A}} \mathbb{I}_T(\mathcal{A}|x_t, b_{0:t-1}).$$

Intuition: interactivity is determined by the dependence between the past and future behaviour of the automaton. This behaviour is determined by a substate-space that grows with the horizon ($T$). Specifically, the future behaviour of an embedded automaton is determined by (i) the universal-local environment's transition function, (ii) the embedded automaton's initial internal state, and (iii) a substate-space of the environment that grows with the horizon of behaviour considered. If an embedded automaton's past behaviour were predictive of its future behaviour of a given horizon, then it would also imply that its past behaviour is predictive of a substate-space growing with that horizon. For a large enough horizon, the size of this substate-space will eventually be larger than the capacity of the automaton. An embedded automaton with a given capacity cannot maximize interactivity beyond a given horizon, meaning that it actively faces an implicit capacity constraint.

Formally: an automaton with sufficiently high interactivity will produce a behaviour with high unconditional Kolmogorov complexity, $\mathbb{K}(b_{t:t+T-1}) + O(1) > C(\mathcal{A})$, where $O(1)$ is a constant independent of the automaton. Because the behaviour is generated by the automaton, we know that the Kolmogorov complexity is also upper bounded by the capacity, $C(\mathcal{A}) + O(1) > \mathbb{K}(b_{t:t+T-1})$.

Thus, we can upper bounded interactivity in terms of the unconditional complexity,

$$\mathbb{I}_T(\mathcal{A}|x_t, b_{0:t-1}) = (\mathbb{K}(b_{t:t+T-1}) - \mathbb{K}(b_{t:t+T-1} \,|\, b_{0:t-1}))$$
$$\leq \mathbb{K}(b_{t:t+T-1})$$

Which uses the fact that the conditional algorithmic complexity is positive, $\mathbb{K}(b_{t:t+T-1} \,|\, b_{0:t-1}) > 0$.

Next we use the fact that the Kolmogorov complexity of a sequence produced by an automaton is upper bounded by its capacity, which implies that,

$$\max_{\mathcal{A}} \mathbb{I}_T(\mathcal{A}|x_t, b_{0:t-1}) \leq \mathbb{K}(b_{t:t+T-1}) \leq C(\mathcal{A}) + O(1)$$

**Lower Bounding Interactivity By Capacity**

For the lower bound, we provide a constructive proof in which we show how an automaton can scale a lower bound on its interactivity proportional to its capacity.

In particular, the automaton will use a brute-force method which simulates a much smaller automaton along with a stored history. That is, a subroutine will simulate an automaton with capacity $\alpha C(\mathcal{A})$, where $\alpha < 1$. The automaton will use its remaining capacity to store on the order of $(1 - \alpha)C(\mathcal{A})$ of the previous input-output pairs. It then runs an enumeration strategy as a subroutine, which simulates the smaller automaton with capacity $\alpha C(\mathcal{A})$. Specifically, the subroutine enumerates over all possible $T$-horizon futures and selects the next output that maximizes the resulting $T$ horizon unconditional complexity and minimizes the $T$-horizon conditional complexity with respect to the stored history of length $(1 - \alpha)C(\mathcal{A})$. While this is computationally expensive, we can choose $\alpha$ to be small enough that this is possible. The interactivity achieved in this setting is thus, $\alpha C(\mathcal{A}) - O(!)$

Putting these together, we have the lower and upper bounds of:

$$\alpha C(\mathcal{A}) - O(1) < \max_{\mathcal{A}} \mathbb{I}_T(\mathcal{A}|x_t, b_{0:t-1}) \leq \mathbb{K}(b_{t:t+T-1}) \leq C(\mathcal{A}) + O(1)$$

$\square$

*Proof of Theorem 2.* We provide a proof for each of the two desiderata

*(i)* The first property follows from an argument that is similar to Theorem 1, but adapted to a learning agent, $\mathcal{A}$, with some bounded capacity $C(\mathcal{A})$. We are interested in what interactivity the best such agent can achieve, $\max_{\mathcal{A}} \mathbb{I}_T(\mathcal{A}|x_t, b_{0:t-1})$.

A bounded agent that maximizes its interactivity will have a non-zero unconditional agent-relativized complexity, $\mathbb{K}_{\mathcal{A}}(b_{t:t+T-1}|\epsilon) > 0$ (otherwise, its interactivity would be zero). This implies that the unconditional Kolmogorov complexity of its behaviour is on the order of the the capacity of the agent, $\mathbb{K}(b_{t:t+T-1}) \geq C(\mathcal{A}) - O(1)$, where $O(1)$ is a constant independent of the agent. Because the behaviour is generated by the automaton, we know that the Kolmogorov complexity is also upper bounded in terms of the capacity, $C(\mathcal{A}) + O(1) \geq \mathbb{K}(b_{t:t+T-1})$.

Such an agent will also have low conditional agent-relativized complexity (otherwise, its interactivity would be low). An optimal learning agent that minimizes the agent-relativized complexity, $\mathbb{K}_{\mathcal{A}}(b_{t:t+T-1}|b_{0:t-1}) = 0$, has conditional Kolmogorov complexity is strictly less than the capacity of the agent, $\mathbb{K}(b_{t:t+T-1}|b_{0:t-1}) < C(\mathcal{A})$. In fact, we have, for $\alpha < 1$, that $\mathbb{K}(b_{t:t+T-1}|b_{0:t-1}) \leq \alpha C(\mathcal{A})$. This is because the agent can only use a fraction of its capacity on predicting its future behaviour (in addition to making predictions, an agent selects actions, and updates its substate).

Taken together, we have that the performance of an interactivity-seeking agent interactivity is upper and lower bounded by capacity,

$$(1 - \alpha)C(\mathcal{A}) - O(1) \leq \max_{\mathcal{A}} \mathbb{I}_T(\mathcal{A}|x_t, b_{0:t-1}) \leq C(\mathcal{A}) + O(1).$$

An agent with a given capacity cannot maximize its interactivity without increasing its capacity. Thus, a bounded agent that seeks to maximize its interactivity through learning is limited by its finite capacity constraint.

*(ii)* For the second property, we demonstrate the necessity of continual adaptation for maximizing interactivity, by considering the role of the embedded agent's transition function.

**Continual Adaptation in Automata**

First we consider the finite-state automaton, $\mathcal{A}$, and how its substate transition function, $u$, encodes its learning. An automaton agent that has stopped learning is thus equivalent to one that stops updating its internal state. In this case, the automaton's internal state remains constant $z_{t'} = z$ for all $t' > t$. A finite-state automaton has a capacity on the order of $C(\mathcal{A}) \in O(poly(|A|))$. But, a finite-state automaton that does not update its internal state, denoted by $\mathcal{A}^-$, has a reduction in its capacity. In particular, the capacity is reduced to $C(\mathcal{A}^-) = O(|\Omega|_X|)$, because the terms needed to encode the transition function are no longer needed for an automaton that does not use the transition function. Using the upper bounds on interactivity from the Theorem 1, we conclude that an agent that stops learning reduces its future output complexity from $O(|I_A||A| \log |A|)$ to $O(|I_A|)$. Thus, it is suboptimal to stop learning.

**Continual Adaptation in Reinforcement Learning Agents**

**Value parameters**: If the parameters of the value function stop being updated, the interactivity objective immediately collapses to 0.

**Policy parameters**: Suppose the policy's parameters stop being updated, meaning that the policy becomes fixed. Then the sequence of actions taken by the fixed policy becomes a Markov process, which is predictable. Under the Markovian dynamics induced by a fixed policy, the value function could converge to an optimal static prediction of the future behaviour. This would lower both the unconditional Kolmogorov complexity and interactivity. Thus, an agent maximizing interactivity is suboptimal if it stops updating its policy parameters.

□

## C Additional Background and Related Work

### C.1 Additional constraints on embedded agents

There are two additional ways in which an embedded automaton is implicitly constrained:

1. **Minimum size**: The size of an embedded automaton, including the size of its input and output spaces, cannot be arbitrarily small, and thus there exists a minimum size. This implies that the automaton cannot read and write to arbitrarily small parts of the environment, constraining its observation and action spaces.

2. **Simulation time**: Simulating an embedded automaton in a universal-local environment may also incur a simulation overhead. This constrains the automaton by the fact that several transitions in the environment may be necessary to simulate a single transition for the automaton.

While the embedded automaton is computationally constrained relative to its environment, these two additional constraints limit the information made available to the automaton about the environment. Specifically, an automaton generally cannot observe, process and output information at the same granularity, or at the same timescale, as the environment because of constraints on its size and its simulation time.

### C.2 Algorithmic complexity

The Kolmogorov complexity (Kolmogorov, 1965; Solomonoff, 1964; Chaitin, 1966) of an object (encoded as a binary string) is the length of the shortest program that computes it and halts. Unlike traditional information theory, it measures the complexity of an individual object without depending on a stochastic source or ensemble.

The Kolmogorov complexity of a string depends on the choice of a universal Turing machine. However, since any universal Turing machine can simulate another (e.g., via a compiler), the choice of the machine affects the Kolmogorov complexity by, at most, an additive constant independent of the specific string (Li and Vitányi, 2019).

Kolmogorov complexity is closely tied to compression, where the shortest description represents the most efficient compression for the given universal Turing machine. Although Kolmogorov complexity is uncomputable, it is possible to compute improving upper bounds by searching over all possible programs in parallel and tracking the shortest candidate that generates the target string (Li and Vitányi, 2019).

### C.3 AIXI

AIXI defines a general Bayes-optimal reinforcement learning agent in an unknown computable environment (Hutter, 2005). In this framework, the environment is represented by a Turing machine with unidirectional input and output tapes, and bidirectional working/internal tapes. The agent's actions are received by the environment on its input tape, based on which it can write a computable history-based reward and observation on its output tape.

The AIXI agent acts in a Bayes-optimal manner by planning based on a posterior estimate over all computable environments, using Solomonoff's universal prior as a starting point (Solomonoff, 1964). This prior assigns higher probability to 'simpler' environments–those with lower Kolmogorov complexity. However, both Solomonoff's prior and AIXI are incomputable, making the development of practical approximations within this framework a key area of interest (Veness et al., 2011).

## C.4 Connections to intrinsic motivation and the free energy principle

Previous work has explored several intrinsic drives that can guide agent behaviour without the need for explicit external rewards (Schmidhuber, 2010; Barto, 2013). Many approaches to intrinsic motivation are developed within the framework of traditional RL, where the agents are not constrained relative to the environment. As a result, these approaches may not be well-suited to a big world. Nevertheless, interactivity shares connections to ideas such as mutual information maximization in intrinsic motivation.

The information gain of a dynamics model can serve as an intrinsic or auxiliary reward, promoting curious exploration (Storck et al., 1995; Houthooft et al., 2016) Unlike curiosity driven by information gain, the goal of interactivity is not to learn an accurate model of the world.

Another related concept is Empowerment (Klyubin et al., 2005), where an agent seeks to maximize its control over its environment. Empowerment-seeking agents aim to maximize the mutual information between their actions and future states. Such agents avoid states where their actions have low influence and prefer states that allow for a wide range of controllable outcomes. This objective can also be used to learn a set of behaviours (or options) that lead to different final states (Mohamed and Jimenez Rezende, 2015; Gregor et al., 2016). As discussed earlier, interactivity-maximizing agents produce complex yet predictable behaviour, which is not directly tied to the concept of control. Furthermore, unlike objectives grounded in traditional (Shannon) information theory, interactivity relies on asymmetric algorithmic mutual information between previous inputs and future outputs.

Active inference describes agentic behavior in partially observable environments as the minimization of free energy (Friston et al., 2010; Sajid et al., 2021). Free-energy minimization prefers selecting actions that lead to highly predictable states—inputs that are unsurprising to the agent's model. In contrast to free-energy minimization, maximizing interactivity actively discourages low-complexity predictable states.

## C.5 Relationship to other notions of boundaries and Markov processes

The boundaried Markov process that we define is also similar to other frameworks which introduce explicit boundaries. Active inference, for example, considers a Markov blanket that separates probabilistic nodes in a directed acyclic graph as a way of separating the agent from its environment (Kirchhoff et al., 2018). Open Markov processes have also been defined which have explicit boundary states from which probability can flow in and out of (Baez et al., 2016). Our work applies similar ideas specifically in the computational setting that we consider.

