# OpenReview forum: "The World Is Bigger! A Computationally-Embedded Perspective on the Big World Hypothesis"
_NeurIPS.cc/2025/Conference — NeurIPS 2025 spotlight_

### Official Review · Reviewer_Hb59 · 2025-06-19

**Clarity:** 3
**Significance:** 2
**Originality:** 3
**Rating:** 4
**Confidence:** 2

**Summary:**

This paper introduces a theoretical framework for continual learning based on the big world hypothesis, where agents are implicitly constrained by being embedded within a computationally universal and local environment. The authors formalize such agents as embedded automata and define a new objective called interactivity, which uses Kolmogorov complexity to quantify how much an agent’s future behavior can be both complex and predictable from its past. They prove that any finite-capacity agent maximizing interactivity must continually adapt, as stopping learning is provably suboptimal. To operationalize this, they develop an RL algorithm that approximates interactivity via prediction error and evaluate it on a synthetic behavioral self-prediction benchmark, where an agent learns from its own nonstationary experience stream. The work is mostly theoretical but provides a novel, general perspective on continual adaptation without requiring explicit task boundaries or external rewards.

**Questions:**

1. Your use of universal-local environments (e.g., Conway’s Game of Life) is well-motivated as a constructive proof that embedded agents can exist. However, it’s not entirely clear why locality is essential beyond enabling the embedding. Could you clarify whether locality plays any further role in shaping the agent’s adaptation dynamics or in bounding interactivity?
2. The proposed interactivity measure uses Kolmogorov complexity and is approximated via agent-relative prediction error. How general is this framework in practice? For instance, would interactivity remain a meaningful or tractable objective in high-dimensional or real-world sensory domains (e.g., vision, language)? What kinds of practical approximations might be required?
3. Your behavioral self-prediction benchmark is clean and well-motivated, but it lacks comparisons to continual learning baselines. Can you elaborate on why existing baselines (e.g., EWC, replay buffers) were not included, and what challenges prevent comparison?
4. Your learning algorithm for interactivity involves a form of bi-level optimization (meta-gradients), which is known to be expensive and unstable. Could you clarify whether this is essential to your formulation, or whether simpler approximations (e.g., first-order variants, bootstrapped objectives) might suffice?
5. Your current experiments only include 2-layer networks. What's the rationale behind this choice, and how does the method extend to deeper networks?

**Ethical Concerns:**

["NO or VERY MINOR ethics concerns only"]

**Final Justification:**

Thanks for the detailed response. The explanations and justifications answer my questions/concerns well and I have increased the score,

**Limitations:**

- Practical Scalability:
The paper proposes an elegant but computationally heavy optimization procedure. While the authors mention second-order complexity and meta-gradients, they do not discuss the practical infeasibility of scaling to larger models. A clearer discussion of how the method might break down or require approximation in such settings would be valuable.
- Limited Empirical Validation:
The synthetic benchmark is theoretically motivated but very narrow in scope. The authors should discuss whether their current results demonstrate efficacy beyond toy settings, and that future work is needed to bridge this gap.

**Paper Formatting Concerns:**

The citation style is not typical NeurIPS style.

**Quality:**

2

**Strengths And Weaknesses:**

Strengths:
- Originality:
The paper presents a novel perspective on continual learning grounded in implicit constraints through computational embedding, rather than explicit resource limitations. The formalization of a universal-local environment and the use of Kolmogorov complexity to define the agent’s objective (interactivity) is highly original and conceptually deep.
- Significance:
By framing continual learning in terms of interactivity, the paper offers a general and theoretically motivated setting in which any non-adaptive agent is provably suboptimal. This challenges standard assumptions in RL and provides a new foundation for studying continual adaptation without reliance on task segmentation or explicit forgetting.
- Theoretical Rigor (Quality):
The paper is mathematically precise and includes formal definitions and proofs for all key claims, deferred to a comprehensive appendix. Notably, the results show that interactivity is upper bounded by agent capacity (Theorem 1) and that agents that stop learning are suboptimal (Theorem 2).
- Clarity of Core Ideas:
Despite the abstract nature of the concepts, the paper clearly communicates its main constructions—universal-local environments, embedded automata, and interactivity—with helpful analogies (e.g., Conway’s Game of Life) and visual aids. The contrast with traditional RL and AIXI is particularly helpful for positioning the contribution.

Weaknesses:
- Experiments (Quality / Significance):
The only experiment is a synthetic benchmark (behavioral self-prediction) without comparison to existing continual learning algorithms. While the experiment supports the theoretical claims, its simplicity (max 2 layers) limits practical insight into the applicability or scalability of the framework.
- Clarity of Implementation Details:
Although the theoretical setup is well presented, the reinforcement learning algorithm for maximizing interactivity is described somewhat abstractly. Important aspects like how the inner/outer learning loops interact, and the concrete form of predictions and TD errors, are difficult to follow without significant effort.
- Limited Practical Connection:
The paper is mostly theoretical, and the benchmark does not involve realistic tasks or environments. This makes it unclear how the interactivity framework might inform or improve existing continual learning methods in practice, especially in high-dimensional domains.
- Heavy Theoretical Overhead:
The reliance on Kolmogorov complexity, while elegant, leads to uncomputability in general and requires approximations that introduce complexity and potential instability (as discussed in the limitations). The practical utility of interactivity as a training signal may thus be limited without further simplification.

---

> ### Author Rebuttal · Authors · 2025-07-31
>
> We appreciate your thorough review of our submission. We particularly appreciate you finding the core ideas clear, the overall originality/significance and your attention to detail regarding the implications of our task on scaling.
>
> >Experiments (Quality / Significance): While the experiment supports the theoretical claims, its simplicity (max 2 layers) limits practical insight into the applicability or scalability of the framework.+ Practical Scalability: The paper proposes an elegant but computationally heavy optimization procedure.
>
> We understand that evaluating scalability is a concern of the reviewer and we agree that it is important.
> We would like to point out that our experiments are in a highly novel task that combines online learning, reinforcement learning, meta learning and continual learning. In fact, Reviewer sAho commended us for generating results at all.
>
> In our additional experiments, we found that there is a performance benefit to scaling width and depth of the network. But our proposed task is such that the complexity of the task scales with the agent. This is important from the perspective of continual learning (for if the environment did not have this property, a non-adaptive learner could out-scale it). It does, however, challenge the conventional approach to scaling, as there is a cost: a scaled up agent faces a harder task. We hope that future empirical investigations will map out efficient approaches to scaling using interactivity maximization as a key task for evaluation.
>
> Lastly, we would like to note that the computational scalability of this problem is a feature, not a bug. While it is true that meta-gradients scale particularly poorly, they provide the ground truth Monte Carlo estimates of the dynamics. We anticipate that effective algorithms will need to scale well, and this task particularly stresses the scalability of algorithms to maximize their interactivity.
>
>
> >Clarity of Implementation Details: ... how the inner/outer learning loops interact, and the concrete form of predictions and TD errors
>
> Thank you for bringing up this point. Appendix B included additional implementation details. However, we plan to improve Section 6 to include how it all fits together. The setup is as follows: there is a (meta-) policy and a value function. They are both parameterized with the same architecture (depth/width), and all of their input and output spaces are the same. The value function is updated with TD0 to learn successor features from its experience. This experience is generated by the (meta-)policy in a feed-forward and recursive way ($b_{t+1} = \pi(b | b_t)$). The policy is updated to maximize the difference between (temporal difference errors of the value function while it is learning) and (temporal difference errors of the frozen value function). The meta-gradient comes in because the policy is optimizing TD errors from a value function that is also learning. Thus, TD learning with the value function is the inner loop and the meta-policy maximizing interactivity is the outer loop.
>
> >Limited Practical Connection: The paper is mostly theoretical, and the benchmark does not involve realistic tasks or environments. This makes it unclear how the interactivity framework might inform or improve existing continual learning methods in practice, especially in high-dimensional domains.
>
> The empirical contribution in our paper is: (1) interactivity maximization as an evaluation task for continual learning and (2) the meta reinforcement learning approach to maximizing interactivity. Our experiments serve to demonstrate that maximizing interactivity is a difficult task for current deep learning algorithms.. Our additional results (see shared reply) demonstrate that methods designed to help with continual learning also help in this task, thus providing additional evidence of its utility as a novel continual learning task.
>
> >Heavy Theoretical Overhead: The reliance on Kolmogorov complexity, while elegant, leads to uncomputability in general and requires approximations that introduce complexity and potential instability (as discussed in the limitations). The practical utility of interactivity as a training signal may thus be limited without further simplification.
>
> We use Komogorov complexity as a tool to measure the predictability of an agent’s behaviour sequence. However, we emphasize that Kolmogorov complexity and interactivity are computable for an embedded automaton, and for a bounded agent. Crucially, we show that an agent can approximate this quantity using reinforcement learning.
>
>
> >Your use of universal-local environments (e.g., Conway’s Game of Life) is well-motivated as a constructive proof that embedded agents can exist. However, it’s not entirely clear why locality is essential beyond enabling the embedding. Could you clarify whether locality plays any further role in shaping the agent’s adaptation dynamics or in bounding interactivity?
>
> Thanks for this great question!
>
> First, we emphasize that locality is common to most models of computation, such as the head movement of a Turing machine, and not just the rules of cellular automata.
>
> Locality plays two crucial roles.You are correct about the first role: it is used to define the boundary between the agent and the environment. You also hint at the second role: locality is needed to bound the interactivity objective. Intuitively, locality ensures that short-term dynamics depend on a small portion of the state-space. Without locality, short-term dynamics may depend on arbitrarily larger portions of the state-space, meaning that even a short-term future behaviour sequence can depend on a large portion of the state space. The implication of this is that the Kolmogorov complexity of short behaviour sequences can be arbitrarily large.
>
> >The proposed interactivity measure uses Kolmogorov complexity and is approximated via agent-relative prediction error. How general is this framework in practice? For instance, would interactivity remain a meaningful or tractable objective in high-dimensional or real-world sensory domains (e.g., vision, language)? What kinds of practical approximations might be required?
>
> The reinforcement learning approximation for interactivity can be applied to any agent that learns to make predictions with temporal difference learning, regardless of the environment it is interacting with. We understand that high-dimensional observations can be more difficult to predict than latent states. However, the approximation involves the difference between two future temporal difference errors (one where the agent learns, and one where the agent does not learn). Thus, even if the environment provides high-dimensional observations that are more difficult to predict, the objective rewards any relative improvements in predictability.
>
> >Can you elaborate on why existing baselines (e.g., EWC, replay buffers) were not included, and what challenges prevent comparison?
>
> Our current set of experiments are set up primarily to evaluate the optimization problem underlying maximizing interactivity. While maintaining memory of the past will be important to make further improvements on interactivity, we found that loss of plasticity and general instability to be of far greater importance. In particular, EWC and replay buffers are not well suited for maximizing interactivity. For EWC, there are no discrete tasks: at every timestep the agent updates its policy and creates nonstationarity. And, because of this nonstationarity, old data is not useful for learning. Thus, replay buffers are not effective here as they are in other reinforcement and continual learning experiments.
>
> >Your learning algorithm for interactivity involves a form of bi-level optimization (meta-gradients)...
>
> While meta-gradients is the approach that we take, this is motivated by the fact that meta-gradients provide a Monte Carlo approximation to our objective. More scalable approaches, like first-order gradients, may be important for larger networks, but the full meta-gradient provides an important first step in understanding the dynamics of an agent maximizing interactivity. More sophisticated approaches could work, such as bootstrapped meta-gradients. Viewing interactivity as a value function opens many more sophisticated value approximation algorithms that may improve over Monte Carlo. We anticipate that such progress can be made in future work.
>
> > Your current experiments only include 2-layer networks. What's the rationale behind this choice, and how does the method extend to deeper networks?
>
> While the base network only has 2 layers, the meta-learner has a much deeper effective network size. This is because the meta-gradient for the meta-learner involves a composition as deep as the horizon. That is, the meta-learner has an effective depth of 2*H. This could be better represented by a recurrent architecture, but this would include yet more complexity that we seek to address in future work.
>
>
> >Limited Empirical Validation: The synthetic benchmark is theoretically motivated but very narrow in scope. The authors should discuss whether their current results demonstrate efficacy beyond toy settings, and that future work is needed to bridge this gap.
>
> The behavioural self-prediction task that we introduce may seem toy, but its complexity scales with the function approximator. It can be applied to any learning algorithm that supports an input space equal to its output space. More generally, it could be used as an intrinsic motivation objective but this will require the development of scalable meta-learning algorithms.

---

### Official Review · Reviewer_CJca · 2025-06-22

**Clarity:** 3
**Significance:** 4
**Originality:** 4
**Rating:** 5
**Confidence:** 3

**Summary:**

The paper proposes a framework to think about agent/environment design by considering how the agent is embedded into the environment, and how the agent’s capabilities of computation are inherently limited based on how the agent is embedded in the environment. It proposes an intrinsic measure of interactivity which determines what the agent should be optimizing to learn more about its environment.

**Questions:**

Is the main idea the following: we treat the environment as a computing machine, which contains the agent (automaton), which is also a computing machine. Then since the agent is embedded in the environment, its computation abilities are upper bounded by what the environment allows and what it presents to the agent (partial observability). And then finally, the goal is for the agent to maximize interactivity, which is basically about learning about a future that is maximally complex (since more to learn about) but also you want your past to reduce complexity of future (since it tells us that the past behaviors were “useful”.

**Ethical Concerns:**

["NO or VERY MINOR ethics concerns only"]

**Final Justification:**

This paper is unique and introduces a topic/discussion that I think will bring value to the community. It allows us to step back and think about the problem generally rather than trying to beat SOTA every time.

**Limitations:**

Yes the paper acknowledges the computational limits of the algorithm to maximize interactivity.

**Quality:**

3

**Strengths And Weaknesses:**

Strengths:
- Very unique paper and brings valuable insights.
- The paper is systematic and I think the ideas presented in this paper are deep. It would be a nice addition to the conference.
- I like that the experiments use self-prediction as a loss since it is aligned with intrinsic motivation losses.
- The relaxation to measure complexity conditioned on some horizon H is clever to make this metric more practically useful (Section 5.3)
- I appreciate the measure of interactivity is based on the undiscounted average reward setting, which is also an underappreciated/less studied setting.

Weaknesses
- The empirical results seem a bit rushed. I understand the focus does not have to be on a full empirical analysis, but I think some of the discussion from Appendix B2 should be moved up and the discussion should relate the results to the main thesis of the paper better.
- The surprising finding of instability of linear methods is also not discussed as much, and deserves some more attention.

---

> ### Author Rebuttal · Authors · 2025-07-31
>
> Thank you for your review! We appreciate you referring to the ideas that we develop as deep, as well as your summarization of the main narrative of our paper. We address your questions below:
>
>
> >The empirical results seem a bit rushed. I understand the focus does not have to be on a full empirical analysis, but I think some of the discussion from Appendix B2 should be moved up and the discussion should relate the results to the main thesis of the paper better.
>
> Thanks, we agree that more detail in the main paper would improve clarity. The goal of our paper was to characterize a problem setting that requires continual adaptation. Our characterization was theoretical, and the experiments that we generated were in a fairly novel setting that combines online learning, reinforcement learning, meta learning and continual learning. The space constraint made it challenging to include all the details. In fact, Reviewer sAho commended us for generating results at all.
>
> Moreover, we report additional results under the section titled [EXP1] in our reply to sAho.
>
> >The surprising finding of instability of linear methods is also not discussed as much, and deserves some more attention.
>
> We agree that this is an interesting detail that can use further elaboration. The fact that linear methods collapse in performance is surprising, but one explanation is that even a linear network effectively becomes a deep nonlinear network when learning with meta-gradients. In our updated experiments, we show that learning can be stabilized.
>
> >Is the main idea the following: we treat the environment as a computing machine, which contains the agent (automaton), which is also a computing machine. Then since the agent is embedded in the environment, its computation abilities are upper bounded by what the environment allows and what it presents to the agent (partial observability). And then finally, the goal is for the agent to maximize interactivity, which is basically about learning about a future that is maximally complex (since more to learn about) but also you want your past to reduce complexity of future (since it tells us that the past behaviors were “useful”.
>
> Yes, that is a great summary of the primary narrative in our submission! The three key ideas are the characterization of the universal-local environment, the embedded agent and the relationship between them (the interactivity objective). We also go one step further by taking this characterization and developing a reinforcement learning approach to maximizing interactivity.

---

### Official Review · Reviewer_sAho · 2025-06-26

**Clarity:** 2
**Significance:** 3
**Originality:** 3
**Rating:** 5
**Confidence:** 3

**Summary:**

In "The World Is Bigger: A Computationally-Embedded Perspective on the Big World Hypothesis" the authors propose a new general problem setting for studying (a version of) the continual learning problem. This is achieved first by defining what the author's call a "universal-local environment" into which an agent may be embedded. The embedded agent is explicitly constrained by construction and is shown to be representable as a POMDP. Furthermore, an intrinsic new measure is proposed for assessing the agents ability to adapt to its future environment, and that maximizing this measure is structurally a continual RL problem.

**Questions:**

1. Any idea why "(linear methods) ... always lead to performance collapse"?
2. Interactivity as defined appears to be one sensible measure, were others considered and rejected?

**Ethical Concerns:**

["NO or VERY MINOR ethics concerns only"]

**Final Justification:**

The authors have improved the strength of the paper and clarified some important issues in their rebuttal. The rebuttal responses improve the clarity and justification for some of the choices made in the paper, and improved the experimental coverage to a level more suitable for a theoretical contribution. I'm increasing my score from a 4-boarderline, to a 5-accept.

**Limitations:**

yes

**Quality:**

3

**Strengths And Weaknesses:**

Strengths:
* The paper presents what I would consider to be a bold contribution to the continual learning area
* The paper presents a valuable extension (in spirit at least) to AIXI, which was an important conceptual contribution to the field
* The formalism for much of the work in the continual learning set up is ad hoc, and this is a contribution towards placing those efforts on more stable grounds
* The paper's motivation and writing are clear (if perhaps its technical presentation might be improved slightly)
* The authors are to be commended for generating *results* from their construction (i.e. presenting an end-to-end utilization of the setup) with all that that entails


Weaknesses:
* Beyond an abstract theoretical construction it is not immediately clear to me if this will be of practical use to the community. Beyond the recovery of the "common desideratum of the continual learning problem"(which indeed is likely by explicit construction) it's not clear that studying agents/algorithms in this setting will provide insight to real-world systems
* There are a lot of contributions here: the idea of "universal-local" environments, the definition of boundaries Markov processes, how to embed agents therein, demonstrating that this embedded construction has the desired properties and representations, the introduction of a suitable objective measure, and how that may be solved with TD methods etc., but with all that said, the empirical results feel almost incidental, and indeed weakly supported.
* Minor typos here and there:
> typo: line 76: "Interactivity (as) viewed..." -or- "Interactivity (when) ~can be~ viewed from..."
> typo: line 325 "... algorithms that stops learning..."
> type: line 327 "We trained two a linear network and a non-linear network..." (trained two what?)
> typo: line 352 "... is not be appropriate..."

---

> ### Author Rebuttal · Authors · 2025-07-31
>
> We would like to thank the reviewer for their insightful comments and questions. We particularly appreciate you referring to our work as a bold contribution, and your recognition of how our work builds on and expands the foundation for continual learning. We have addressed your concerns below. And, we also use some of this rebuttal for additional details regarding our experiments.
>
> >Beyond an abstract theoretical construction it is not immediately clear to me if this will be of practical use to the community. Beyond the recovery of the "common desideratum of the continual learning problem"(which indeed is likely by explicit construction) it's not clear that studying agents/algorithms in this setting will provide insight to real-world systems
>
> While our approach is more on the abstract side, the behavioural self-prediction task provides a new way to evaluate an algorithm’s capability for continual learning. That is, rather than the current standard practice in continual learning where non-stationarities are designed, our proposed task evaluates a learning algorithm's ability to create its own nonstationarity. This is significant because it captures an important aspect of real world data, which  is often nonstationary yet predictable.
>
> >There are a lot of contributions here: the idea of "universal-local" environments, the definition of boundaries Markov processes, how to embed agents therein, demonstrating that this embedded construction has the desired properties and representations, the introduction of a suitable objective measure, and how that may be solved with TD methods etc., but with all that said, the empirical results feel almost incidental, and indeed weakly supported.
>
> Please see our shared reply on the experiments, including a description of new results.
>
> >Minor typos here and there:
>
>
> Thank you for catching these!
>
> > Any idea why "(linear methods) ... always lead to performance collapse"?
>
> Thanks for bringing this up, it is a subtle point that we want to discuss further. The fact that linear methods collapse in performance is surprising compared to the stability of linear methods in reinforcement learning and continual learning. One explanation is that learning with meta-gradients and a linear network is similar to learning with gradients and a deep nonlinear network. In our shared reply we discuss this point further, and strategies we found effective in stabilizing learning.
>
> > Interactivity as defined appears to be one sensible measure, were others considered and rejected?
>
> There are three levels to this question:
> - Our starting point was to measure the “predictable complexity” of future behaviour, and the motivation behind this was to characterize adaptive behaviour.
> - Kolmogorov complexity-based interactivity for finite automata is one representation of this idea. An alternative that we considered is algorithmic/Solomonoff probability of a given sequence, which can be defined on the environment as the log of the number of environment configurations that produce that sequence. The advantage of this approach is that, once defined with respect to our environment, it provides better intuition and can be visualized. The disadvantage of this approach is that it is not as well-known as Kolmogorov complexity, and would require more space for elaboration.
> - Finally, in the approximation to interactivity using reinforcement learning, we used an average reward formulation. However, we also considered a discounted approach. The advantage of discounting is that it is more familiar to RL researchers, and there are more well-known algorithms for the discounted settings. The disadvantage is that discounting would introduce more approximation, given that interactivity was defined without discounting.
>
>
> ### [EXP1] We also included additional results with activation functions that have been found to be stable for continual learning:
>
> We found that ReLU activations struggle to sustain their interactivity. While linear activations are not able to achieve as high interactivity, they are relatively stable. CReLU and Tanh seem to balance stability and performance, whereas a sinusoid is the least stable and was the only activation function to diverge.
>
> | Method | Iteration 2500 | Iteration 5000 | Iteration 7500 | Iteration 10000 |
> |---|---|---|---|---|
> | deep linear | 0.017 (0.002) | 0.016 (0.003) | 0.014 (0.002) | 0.019 (0.004) |
> | relu | 0.012 (0.003) | 0.017 (0.005) | 0.012 (0.003) | 0.008 (0.002) |
> | leaky | 0.015 (0.004) | 0.024 (0.007) | 0.030 (0.008) | 0.010 (0.003) |
> | sin | 0.017 (0.008) | 0.029 (0.009) | 0.016 (0.006) | 0.007 (0.006) |
> | crelu | 0.012 (0.003) | 0.010 (0.002) | 0.011 (0.002) | 0.019 (0.005) |
> | tanh | 0.017 (0.003) | 0.021 (0.005) | 0.021 (0.005) | 0.022 (0.006) |
>
> The following results include double the number of iterations with layer norm. We found that higher interactivity could be achieved with layer norm. In continual learning, layer norm is helpful in ensuring that activations stay active throughout training. However, nonstationarity with layer norm can also have the effect of slowly lowering the effective learning rate and thus slowing learning. You can see that initial performance decreases over time for most nonlinear activation functions, indicating that the initial benefit of layer norm diminishes due to the nonstationarity.
>
> | Method | Iteration 5000 | Iteration 10000 | Iteration 15000 | Iteration 20000 |
> |---|---|---|---|---|
> | deep linear + LN | 0.057 (0.012) | 0.047 (0.010) | 0.034 (0.008) | 0.055 (0.015) |
> | relu + LN | 0.062 (0.010) | 0.046 (0.009) | 0.028 (0.006) | 0.033 (0.008) |
> | leaky + LN| 0.071 (0.012) | 0.050 (0.009) | 0.038 (0.007) | 0.039 (0.007) |
> | sin + LN | 0.062 (0.010) | 0.025 (0.005) |
> | crelu + LN| 0.077 (0.013) | 0.047 (0.010) | 0.030 (0.006) | 0.032 (0.006) |
> | tanh + LN| 0.109 (0.020) | 0.055 (0.012) | 0.045 (0.011) | 0.060 (0.014) |

---

> > ### Comment · Reviewer_sAho · 2025-08-06
> > **Reply to rebuttal**
> >
> > I would like to thank the authors for their responses here and engagement in the review process.
> >
> > As a few reviewers have pointed out, despite the abstract nature of the contribution some more work in the experimental results would strengthen this submission. The response the authors provided here (and to a related question) regarding the surprising collapse of linear methods is interesting and should be included in the final experimental discussion along with the results shared here.
> >
> > To somewhat clarify one of my own critiques to which the authors responded:
> > ""
> > While our approach is more on the abstract side, the behavioural self-prediction task provides a new way to evaluate an algorithm’s capability for continual learning. That is, rather than the current standard practice in continual learning where non-stationarities are designed, our proposed task evaluates a learning algorithm's ability to create its own nonstationarity. This is significant because it captures an important aspect of real world data, which is often nonstationary yet predictable.
> > ""
> > while I am appreciative of the value of the self-prediction task, assumedly we would at some point seek to develop good continual learning algorithms (as measured in this domain with self-prediction) and have them perform well on real-world task-streams. It is not immediately clear that this would be the case - i.e. where non-stationary by predictable meets "continual" in the real world.

---

> > > ### Author Response · Authors · 2025-08-07
> > >
> > > I appreciate the reviewers comments on this topic. We will  include our updated results and expanded discussion of our experiments in section 6 of the main paper.
> > >
> > >
> > > >we would at some point seek to develop good continual learning algorithms (as measured in this domain with self-prediction) and have them perform well on real-world task-streams. It is not immediately clear that this would be the case
> > >
> > > The real world provides many different examples of nonstationarity depending on the task. For example, nonstationarities can arise from wear-and-tear in physical systems, from several agents interacting with each other (such as in financial markets), and high dimensional dynamical systems (weather). In principle, these systems are predictable and, with more information/memory and more computation for processing, predictions could be improved. If the system were truly unpredictable, then such a system would not be of interest for learning. This means that two properties of real-world nonstationarity include: (1) continual adaptation is necessary for making accurate predictions, and (2) predictive performance can be improved with more memory and computation.
> > >
> > > Our work and the proposed task characterizes this general form of nonstationarity common to real-world tasks, in which an agent experiences apparent nonstationarity that can be mitigated with more memory and computation. As our theoretical and empirical results show, performance on this task requires continual adaptation and this performance can be improved by scaling up the agent. Furthermore, our approach provides a computational framework for the interaction loop between a physical agent and the physical world.
> > >
> > > Because our task is designed to evaluate agents for their scalability in continually adaptation, we are directly incentivizing the design of algorithms that would be effective to deal with real-world nonstationarity. Such scalable and adaptive agents are critical for addressing the challenges of the real-world, and our work provides a first step in in this direction.

---

### Official Review · Reviewer_X3Zz · 2025-07-03

**Clarity:** 2
**Significance:** 2
**Originality:** 3
**Rating:** 4
**Confidence:** 3

**Summary:**

Authors define a class of environments, "universal Markov environments" equivalent to universal Turing machines, and a "locality" condition. Authors then formalize agents/automata within these environments, and define interactivity as a complexity-theoretic measurement of the ability to adapt future behavior to past experience. Authors further define agent-relativized complexity and show they can approximate this with a value function trained with RL TD learning. Authors prove that agents with the goal of maximize interactivity necessarily employ continual learning; stopping learning is necessarily suboptimal. Authors implement this method on an environment.

**Questions:**

- Which definitions have already been defined in the literature, and which parts are novel contributions?
- Section 1 states that "Interactivity is similar to previously considered intrinsic motivation objectives 70 (Chentanez et al., 2004; Schmidhuber, 2010), and specifically predictive information (Bialek et al., 71 2001; Still and Precup, 2012). However, interactivity differs because of its formulation in terms 72 of behaviours using Kolmogorov complexity. This makes interactivity better suited to sequential 73 decision making in the constrained and partially observable setting that we consider." Is there a reason predictive information is less-well suited to sequential decision making? I would have thought that interactivity being defined based on Kolmogorov complexity would make it _worse_ for decision making settings.
- I would explicitly define "capacity".
- Is RL the most natural technique to learn in these environments?

**Ethical Concerns:**

["NO or VERY MINOR ethics concerns only"]

**Final Justification:**

The authors addressed my questions about the use of Kolmogorov complexity and interactivity, as well as my concern about the experiment. Because of this, I will raise my rating to 4.

**Limitations:**

- Interactivity can only be measured through an approximation, the utility of which is unclear.
- Experiments showing the usefulness of these measures are not robust.

**Paper Formatting Concerns:**

- Typo in Section 7, "characterizing the implicit constrains"

**Quality:**

2

**Strengths And Weaknesses:**

Strengths
- The idea of universal-local environments is interesting, and I like the example of Conway's Game of Life. I like the idea of having a general computational framework that can describe agency in terms of complexity.

Weaknesses
- I'm unsure if Kolmogorov complexity is the right complexity measure to use to describe agency- it may be, but it is also very general and uncomputable. It's unclear if the approximation the authors define is useful for applications.
- I don't think "implicitly constrained" is defined sufficiently.
- I'm not sure the authors explain why interactivity is a measure that should be optimized, beyond that it is a measure that requires continual learning by Theorem 2.
- Weak experiments: There is just one set of experiments in Figure 4, and they are not described in Section 6 in sufficient detail. For example, the nature of the environment is not described.

---

> ### Author Rebuttal · Authors · 2025-07-31
>
> We are glad to hear that the reviewer found our submission interesting, and we thank them for going through our paper in detail. We address your individual questions below. Please also see our shared reply for more details regarding our experiments.
>
>
> >I'm unsure if Kolmogorov complexity is the right complexity measure to use to describe agency- it may be, but it is also very general and uncomputable. It's unclear if the approximation the authors define is useful for applications.
>
> Thanks for the opportunity to clarify. To be clear, the idea behind interactivity is to characterize agency in terms of future behaviour that is predictable from past behaviour. We use Komogorov complexity as a tool to measure the predictability of an agent’s behaviour sequence. You’re correct to point out that this is a general approach, like previous work at the intersection of algorithmic information theory and AI (AIXI), and we see this generality as a strength. We also would like to emphasize that Kolmogorov complexity and interactivity is computable for an embedded automaton, and for a bounded agent. Crucially, we show that an agent can approximate this quantity using reinforcement learning.
>
> >I don't think "implicitly constrained" is defined sufficiently.
>
> Thanks for raising this point. We will include an explicit discussion of “implicitly constrained” in the paper. Briefly, implicit constraints refers to constraints that are not explicitly imposed in the agent's objective (e.g., autonomous vehicles routing passengers have a constraint in their objective to follow traffic rules). Rather, these constraints emerge between the agent and its relationship to the environment. For example, a robot with a broken sensor is implicitly constrained by that broken sensor. This constraint is not reflected in its objective or algorithm, but can be detected by the agent through its now limited ability to sense the world.
>
> >I'm not sure the authors explain why interactivity is a measure that should be optimized, beyond that it is a measure that requires continual learning by Theorem 2.
>
> We first motivated interactivity as a measure of an agent’s ability to learn from experience:
>
> [line 199-200]: An agent’s capability for learning can be characterized by its ability to adapt its future behaviour using its past experience. We propose interactivity to measure an embedded agent’s intrinsic ability to adapt its future behaviour, towards higher complexity, conditioned on its past behaviour.
>
> Agents with high interactivity are those that use their past experience to predict and then adapt their future behaviour. We position this as a continual learning task, but  incentivizing this type of adaptive behaviour could be useful for designing reinforcement learning agents that interact with their environment to discover what can and cannot be predicted or controlled. Our current work is the first step in understanding the challenges in this objective, and we are excited at the prospect of interactivity as an RL auxiliary objective in future work.
>
> >Weak experiments: There is just one set of experiments in Figure 4, and they are not described in Section 6 in sufficient detail. For example, the nature of the environment is not described.
>
> We would like to point out that our experiments are in a highly novel task that combines online learning, reinforcement learning, meta learning and continual learning. In fact, Reviewer sAho commended us for generating results at all. Moreover, additional details regarding the experiments were included in Appendix B.
>
> We have also added additional results which you can find under [EXP1] (Due to space, it is in another comment).
>
> >Which definitions have already been defined in the literature, and which parts are novel contributions?
>
> We will make the following more clear in our submission: Definitions 1 and 2 have been defined in the literature, and the remaining are novel contributions. Definitions 3,4 and 5 develop the key tools for characterizing the universal-local environment that we consider. Definition 6 is similar to the standard automaton, but defined on our universal-local environment. Definition 7 is similar to the standard conditional Kolmogorov complexity, but adapted to the automaton embedded in the universal-local environment.
>
> >Section 1 states that "Interactivity is similar to previously considered intrinsic motivation objectives 70 (Chentanez et al., 2004; Schmidhuber, 2010), and specifically predictive information (Bialek et al., 71 2001; Still and Precup, 2012). However, interactivity differs because of its formulation in terms 72 of behaviours using Kolmogorov complexity. This makes interactivity better suited to sequential 73 decision making in the constrained and partially observable setting that we consider." Is there a reason predictive information is less-well suited to sequential decision making? I would have thought that interactivity being defined based on Kolmogorov complexity would make it worse for decision making settings.
>
> This is a great question, and we appreciate the opportunity to elaborate. Predictive information, and other notions from Shannon entropy, are defined for stationary probability distributions. In our setting, we do not assume any stationary probability distribution, as this is not always well-defined in a computable environment [1]. Kolmogorov complexity, and other tools from algorithmic information theory, are better suited to this setting as they are defined for sequences, rather than probability distributions. There is a long history of using this approach for understanding the limits of AI agents [2].
>
> >I would explicitly define "capacity"
>
> Thank you for the suggestion! In our computational framework, capacity is measured by the size of the automaton’s state-space. We will clarify this in our submission, as this is an important point. The size of an automaton’s state-space fundamentally bounds the complexity of the behaviour that can be produced.
>
> >Is RL the most natural technique to learn in these environments?
>
> There are three components that make reinforcement learning a natural fit: 1) the environment is a partially-observable Markov process, 2) the agent’s objective involves a long-term prediction, like a value function, 3) the agent’s performance depends on its behaviour sequence, meaning that it needs to explore and learn through trial-and-error. For these reasons, reinforcement learning provides the right problem characterization, as well as algorithms as we demonstrate in our approximation.
>
> >Interactivity can only be measured through an approximation, the utility of which is unclear.
>
> We want to be clear that this is largely true for several other quantities in machine learning, such as optimal value functions and variational lower bounds. Interactivity is a value function, and we generate Monte-Carlo estimates in our experiments just as is done in calculating the return in policy gradient. In addition, our approximation maximizes a lower bound, which is a theoretically sound approach to optimization, like in variational lower bounds.
>
> >Experiments showing the usefulness of these measures are not robust.
>
> We want to highlight that our results are not about measures being robust, as our results show the performance of conventional algorithms on our proposed task (maximizing interactivity). Our results simply highlight the difficulty of applying standard learning algorithms, showing that they are unable to sustain learning continually.
>
>
> [1] Müller, Markus. "Stationary algorithmic probability." Theoretical Computer Science 411.1 (2010).
>
> [2] Hutter, Marcus. "A theory of universal artificial intelligence based on algorithmic complexity." arXiv preprint cs/0004001 (2000).

---

> > ### Comment · Reviewer_X3Zz · 2025-08-05
> >
> > Thank you for your comments. I believe the authors have addressed my concerns well, and I will increase my rating.

---

> > > ### Author Response · Authors · 2025-08-07
> > >
> > > We would like to thank the reviewer for their acknowledgement and for considering a rating increase. Please let us know if you have any additional questions.

---

### Author Response · Authors · 2025-08-02
**Shared Reply**

We would like to thank the reviewers for their constructive feedback and questions. The following strengths were shared in the reviews for our submission:

- Highly original and interesting: All reviewers praise the novel theoretical framework combining universal-local environments, embedded automata, and our interactivity objective, which measures the adaptivity of an agent’s behaviour using Kolmogorov complexity. [X3Zz, sAho, CJca, Hb59]
- Conceptual/Theoretical Depth: The majority of the reviewers noted the strong computational foundation that places continual learning on more theoretical grounds. In particular, showing that interactivity can be approximated by reinforcement learning, and that non-adaptive agents are suboptimal for maximizing interactivity. [sAho, CJca, Hb59]
- Overall clarity: While some commented that there is a  “theoretical overload” in our presentation, the reviewers understood the core idea of our paper [sAho,CJca, Hb59].

We also answer questions and comments were shared by the reviewers around our experiments

### Additional details and results

We will update our submission to present a more detailed analysis of the linear baseline, as it provides a key insight into the dynamics of the proposed task.

Specifically, we consider the case of a linear predictor learning successor features of its observations with TD0, where the observations are generated by a linear meta-policy maximizing interactivity. The meta-policy generates rollouts that both (1) minimizes the TD errors of the dynamic predictor that learns on the rollout and (2) maximizes the TD errors of the static predictor that stays fixed on the rollout. The meta-policy thus learns to generate observations for which continual learning is optimal.

In our updated results, we achieved a stable linear baseline by normalizing the outputs of the meta policy so that the generated experience stream is bounded. Specifically, we used a stop-gradient on the normalization terms so that, for a linear meta-policy, the meta-gradient dynamics are linear. Without this stop-gradient output normalization, the meta policy would have nonlinear dynamics that incentivize observations with larger magnitudes to maximize the TD errors of the static predictor.

Next, we experimented with varying the meta-policy architecture that generates 1024 dimensional observations for the linear predictor. We considered a deep linear meta-policy and a deep ReLU meta-policy with varying widths and depths. Our result table indicates that only the deep linear meta-policy is capable of sustaining interactivity, and this increases with greater widths/depths.


| Depth-> |2|3|4|
|-|-|-|-|
|Linear|0.016 (0.003)|0.10 (0.005 |0.12 (0.006)|
|ReLU|0.004 (0.003)|0.01 (0.004)|0.001 (0.002)|

|Width->|128|256|512|1024|
|-|-|-|-|-|
|Linear|0.036 (0.001)|0.037 (0.001)|0.10 (0.005)|0.13 (0.007)|
|ReLU|0.006 (0.002)|0.004 (0.003)|0.01 (0.004)|0.003 (0.004)|

Finally, we comment on the qualitative behaviour of a highly interactive meta-policy. The deep linear meta-policy learned to generate wave-like observations which vary in phase and amplitude over the course of training, without any inductive bias but interactivity maximization. The dynamic linear predictor is capable of adapting locally to the wave-like observations, but the static linear predictor is incapable of learning a globally accurate prediction because waves are nonlinear. The deep ReLU networks could only sustain generating wave-like observations for a transient period before the output would collapse to a constant, indicating loss of plasticity and resulting in low interactivity.

### Common methods for continual learning (e.g., EWC, replay buffers)

We aimed to understand the performance of basic algorithms as a first step due to the unique nature of our proposed task which combines online, reinforcement, meta and continual learning. In particular, EWC assumes a sequence of discrete tasks, whereas behavioural self-prediction is one continually changing task. In addition, a replay buffer would not be effective because of the nonstationarity making past data stale. Furthermore, our algorithm can be viewed as “on-policy" due to its use of rollouts (similar to PPO) which is another reason replay buffers are not applicable here.

### Overall Applicability
We presented behavioural self-prediction as a proof-of-concept task for evaluating an algorithms capability for continual learning outside of any particular environment. The task scales with the complexity of the agent and an agent that stops learning is suboptimal. This makes it so that the need for continual learning cannot be outscaled, and even larger networks must continually adapt. Interactivity could also be used as an intrinsic motivation objective to incentivize RL agents to produce adaptive behaviour. However, we found that maximizing interactivity in the behavioural self-prediction task is challenging for current learning algorithms.

---

### Note · Authors · 2025-08-13

We would like to take this opportunity to once again thank the reviewers. Before the rebuttal, all reviewers expressed positivity regarding the core contribution of our work:

- Reviewer X3Zz: "I like the idea of having a general computational framework that can describe agency in terms of complexity."

- Reviewer sAho: "a bold contribution to the continual learning area [...] a valuable extension [...] this is a contribution towards placing [previous] efforts on more stable grounds"

- Reviewer CJca: "Very unique paper and brings valuable insights. [...] I think the ideas presented in this paper are deep. It would be a nice addition to the conference."

- Reviewer Hb59: "provides a new foundation for studying continual adaptation without reliance on task segmentation or explicit forgetting."

In our author-reviewer discussion, we answered followup question which we briefly summarize below.

- Reviewer X3Zz: We clarified the distinction between implicit vs explicit constraints, as well as the motivation for algorithmic information, rather than Shannon information, as a tool to characterize sequences with apparent nonstationarity. The reviewer stated that their concerns were addressed and that they would increase their score, but have not yet updated their review to reflect our discussion.

- Reviewer sAho: We provided additional details regarding how the nonstationary produced by the agent in the behavioural self-prediction task relates to nonstationarity in the real world, and the reviewer acknowledged our rebuttal.

- Reviewer CJca: The reviewer recommended acceptance of our paper, and acknowledged our rebuttal which pointed to additional experimental results.

- Reviewer Hb59: We clarified how the behavioural self-prediction task provides a novel evaluation of an algorithm's scalability and continual adaptability, how current algorithms struggle to learn continually, and provided additional empirical evidence. The reviewer acknowledged our rebuttal.

Lastly, we provided a more thorough analysis of the behavioural self-prediction task, achieving stable continual learning performance with the linear baseline, and demonstrated that deep nonlinear networks struggle to sustain interactivity. The gap between the linear baseline and deep nonlinear networks demonstrates the challenges of deep continual learning, and validates behavioural self-prediction as a novel task for evaluating continual adaptation of learning algorithms.

---

### Decision · Program_Chairs · 2025-09-17

**Decision:**

Accept (spotlight)

**Comment:**

The reviewers agree that this work introduces a novel framework for continual learning through universal-local environments and the use of interactivity as a measure. This formalism embeds agents with computational limits and shows that maximizing interactivity naturally leads to continual learning. Reviewers view this as a useful extension of prior work like AIXI, providing a more principled foundation for studying agent-environment interaction.

Strengths noted include the clarity of motivation, systematic treatment of the problem, and an end-to-end demonstration with experiments. The use of Conway’s Game of Life, practical relaxations of complexity measures, and alignment with intrinsic motivation methods were highlighted as novel. I therefore recommend acceptance of this work to NeurIPS and look forward to seeing further research that builds on it.